# United Minds or Isolated Agents? Exploring Coordination of LLMs under Cognitive Load Theory

## Abstract

Large Language Models (LLMs) exhibit a notable performance ceiling on complex, multi-faceted tasks, as they often fail to integrate diverse information or adhere to multiple constraints. We posit that this limitation arises when the demands of a task exceed the LLM's effective cognitive load capacity. This interpretation draws a strong analogy to Cognitive Load Theory (CLT) in cognitive science, which explains similar performance boundaries in the human mind, and is further supported by emerging evidence that reveals LLMs have bounded working memory characteristics. Building upon this CLT-grounded understanding, we introduce *CoThinker*, a novel LLM-based multi-agent architecture designed to mitigate cognitive overload and enhance collaborative problem-solving abilities. *CoThinker* operationalizes CLT principles by distributing intrinsic cognitive load through agent specialization and managing transactional load via structured communication and a collective working memory. We empirically validate *CoThinker* on complex problem-solving tasks and fabricated high cognitive load scenarios, demonstrating improvements over existing multi-agent baselines in solution quality and efficiency. Our analysis reveals characteristic interaction patterns, providing insights into the emergence of collective cognition and effective load management, thus offering a principled approach to overcoming LLM performance ceilings.

## 1 Introduction

The increasing capability of Large Language Models (LLMs) is transforming diverse domains, moving beyond basic text generation towards complex reasoning applications (Chang et al., 2024; Zhao et al., 2024; Li et al., 2024a). Aligning these powerful models with human intent and fostering effective thinking patterns is paramount for unlocking their full potential (Shen et al., 2023). In-Context Learning (ICL) is increasingly employed for this purpose. ICL adapts models via prompt-based guidance without parameter updates (Brown et al., 2020; Lampinen et al., 2024), shaping thinking patterns similarly to finetuning while avoiding retraining (Lin et al., 2024; Zhao et al., 2025; Yang et al., 2024). Its parameter-free flexibility has made ICL a widely adopted approach.

While ICL offers flexibility, it suffers from a performance ceiling when applied to multi-faceted tasks requiring the integration of diverse information sources provided in-context (He et al., 2024; Li et al., 2023b; Kirk et al., 2023). In such scenarios, when guided by extensive in-context information, LLM agents frequently exhibit degeneration of thought, lack of diversity, or an inability to follow multiple requirements (Liang et al., 2023; Huang et al., 2023; Kamoi et al., 2024; Lu et al., 2024). Despite increasing empirical studies on ICL's limitations, the root causes remain underexplored. Concurrently, recent efforts to overcome this ceiling via agent-based solutions have yielded limited success, often relying on heuristics without a firm cognitive grounding (Liu et al., 2023; Zhang et al., 2024c).

To address this gap, we turn to cognitive science for explanatory insight. Similar patterns of performance degradation under high informational demands have long been studied through the framework of Cognitive Load Theory (CLT) (Sweller, 2011; 2003). To better understand this phenomenon in LLMs, we first conduct a pilot study (Section 3). This study provides theoretical grounding by establishing the cognitive load framework for LLM performance limits and empirically examines this analogy through measurable proxies for cognitive load effects. Specifically, following

CLT, we define an agent's **Working Memory (WM)** as its intrinsic, capacity-limited ability to simultaneously hold and process information active in its context, analogous to the capacity of the attention mechanism (Baddeley et al., 1986b). Correspondingly, **Cognitive Load (CL)** is the demand that a task places on an agent's WM, largely determined by the complexity and element interactivity of the information presented via ICL. When the CL imposed by a task exceeds the agent's WM capacity, a state of **cognitive overload** occurs. To examine this WM-CL analogy in LLMs, we conduct experiments measuring attention entropy and perplexity as proxies for cognitive load effects, demonstrating that LLMs exhibit patterns consistent with CLT predictions. Furthermore, recent evidence that LLMs exhibit bounded, human-like WM characteristics (Zhang et al., 2024b; Gong et al., 2024) strengthens our core posit: *The performance ceiling of ICL arises when the demands of the in-context information exceed the LLM's effective cognitive load capacity, mirroring the theoretical limits described by CLT.*

Building on the CLT-grounded understanding and the empirical evidence from our pilot study, we present *CoThinker*, a multi-agent ICL architecture designed to mitigate the cognitive overload imposed by complex tasks. *CoThinker* operationalizes CLT principles through three key functions: (i) dynamic thinking style assignment that adapts to task demands rather than fixed roles, distributing intrinsic cognitive load across specialized agents, (ii) a transactive memory system that maintains shared knowledge about agent expertise and task progress, enabling cognitive offloading and reducing redundant processing, and (iii) a communication moderator that balances cognitive similarity for efficient integration with diversity for comprehensive coverage, creating small-world network properties that minimize coordination overhead while maximizing information flow. In sum, we make the following key contributions:

- We first explain ICL failures on multi-faceted tasks as cognitive overload by formalizing a Cognitive Load Theory-based mapping from human working memory to LLM attention and context limits, and provide empirical evidence.
- We then introduce *CoThinker*, a novel multi-agent ICL architecture that operationalizes Cognitive Load Theory through agent specialization, a transactive memory system, and a moderated communication protocol to mitigate cognitive overload.
- We evaluate *CoThinker*'s effectiveness on complex benchmarks across diverse LLMs, demonstrating its superiority over existing baselines. Additionally, we provide a component analysis that underscores its success in managing cognitive load effectively.

## 2 RELATED WORK

### 2.1 MULTI-AGENT LLM COLLABORATION

The rise of LLMs has spurred research into multi-agent systems (MAS), where LLMs collaborate to tackle complex problems beyond the scope of single agents (Guo et al., 2024; Wang et al., 2024a; Qian et al., 2025). Current approaches include multi-agent debates for idea exchange and critique (Liang et al., 2023; Lu et al., 2024; Wang et al., 2024b; Du et al., 2023), iterative reflection for self-correction (Shinn et al., 2023; Madaan et al., 2023; Yao et al., 2023), and role-playing or functional specialization, where agents divide tasks in complex domains (Li et al., 2023a; Qian et al., 2023a; Hong et al., 2023). Architecturally, research explores optimal communication topologies (Li et al., 2024b), dynamic agent networks (Liu et al., 2023; Wu et al., 2023), mental set diversity (Liu et al., 2025b), and hierarchical coordination (Zhang et al., 2024a). However, these designs often rely on intuition or communication efficiency, with limited grounding in cognitive theories of collaboration and processing constraints (Pan et al., 2025). While recent work on multi-persona self-collaboration (Wang et al., 2023) and meta-prompting (Suzgun & Kalai, 2024) demonstrates empirical benefits from diversity, these approaches lack functional-level understanding of the cognitive mechanisms underlying their effectiveness, operating primarily at the behavioral level without explicit characterization of internal processing states. Our work, *CoThinker*, directly addresses this gap by operationalizing Cognitive Load Theory to enhance collective problem-solving.

### 2.2 LLM FOR HUMAN SIMULATION

The capacity of LLMs to exhibit human-like intelligence (Liu et al., 2025a) and emulate nuanced social behaviors (Zhou* et al., 2024) is foundational to their use as artificial agents. Research has

demonstrated LLMs' ability to simulate human decision-making (Xie et al., 2024), generate believable individual and collective behaviors in social simulations (Chuang et al., 2024a), and adopt distinct personas (Chuang et al., 2024b) Critically, these parallels extend to cognitive characteristics; recent studies suggest LLMs possess bounded working memory and exhibit failure modes under cognitive overload akin to humans (Zhang et al., 2024b; Gong et al., 2024), as discussed in our introduction. Furthermore, interactions between LLM agents can mirror social psychological phenomena (Zhang et al., 2024c; Guo et al., 2024). This confluence of human-like cognitive traits, including limitations and social capabilities, provides a strong rationale for applying principles from human cognitive science—particularly theories like Cognitive Load Theory (CLT) that address cognitive limits—to the design of more effective LLM-based collaborative systems.

# 3 COGNITIVE FOUNDATIONS FOR ENHANCED LLM PERFORMANCE

This section presents our pilot study, which establishes the theoretical foundation for our approach by linking human cognitive limitations to performance ceilings in LLMs. We introduce a cognitive load model based on working memory (WM) analogies (Section 3.1), examine this analogy empirically (Section 3.2), and demonstrate how Cognitive Load Theory (CLT) can address individual limitations and guide LLM system design (Section 3.3).

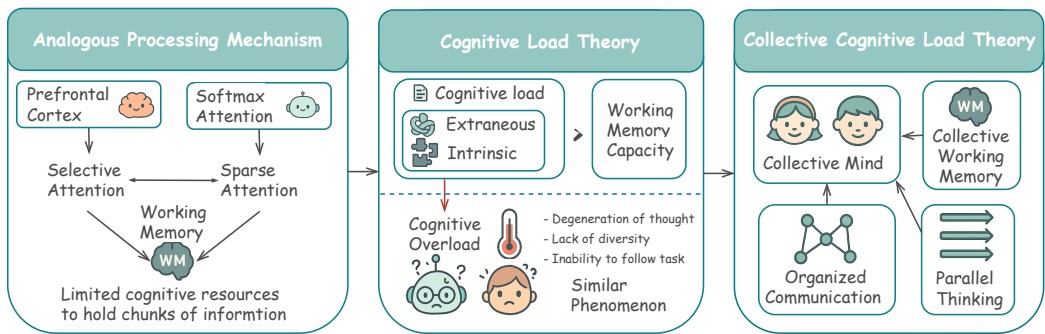

Figure 1: Cognitive Load framework: Using Cognitive Load Theory (CLT), we align human and LLM cognitive load via analogy to explain ICL performance ceilings in complex tasks and to guide methods to mitigate them.

## 3.1 A COGNITIVE LOAD FRAMEWORK FOR LLM PERFORMANCE LIMITS

As depicted in the first and second blocks of Fig. 1, we propose a model that draws parallels between human cognitive limitations and performance ceilings in LLMs.

**Analogous Processing Mechanisms.** Human cognition relies fundamentally on working memory, a capacity-limited cognitive system associated with the prefrontal cortex that employs selective attention to filter and prioritize information during complex cognitive tasks (Baddeley et al., 1986a; Cowan, 2010; Miller et al., 1956). LLMs exhibit intriguing functional parallels through their softmax attention mechanisms, which perform selective focus on input data (Vaswani et al., 2017), with attention heads specializing in distinct processing patterns (Voita et al., 2019). Recent studies provide direct evidence that LLMs possess human-like WM characteristics, exhibiting clear limitations on concurrent information processing with performance degrading predictably as cognitive demands increase (Zhang et al., 2024b; Gong et al., 2024). More discussion in Appendix B.1.1, B.1.2.

**Cognitive Load Theory.** Building upon these working memory analogies, we apply CLT (Sweller et al., 1998; Sweller, 2011) to interpret LLM performance patterns, which are derived from human WM in cognitive science. CLT distinguishes Cognitive Load (CL) between *intrinsic load* (determined by task complexity and element interactivity) and *extraneous load* (arising from instruction presentation). When the combined load exceeds working memory capacity, *cognitive overload* ensues. LLM agents demonstrate analogous performance degradation when tasked with complex problems via In-Context Learning (ICL): tasks requiring extensive multi-step reasoning or integration of numerous constraints can lead to degeneration of thought, lack of diversity, or inability to follow multiple

requirements (Liang et al., 2023; Huang et al., 2023; Kamoi et al., 2024; Lu et al., 2024). We contend that such performance ceilings represent cognitive overload, where total demands surpass the LLM's effective processing capacity. Examples in Appendix B.1.4.

## 3.2 Understanding Cognitive Load and Working Memory in LLMs

We empirically examine the analogy of CL and WM in LLMs. We probe into measurable proxies for cognitive load effects by definition and examine key CLT predictions regarding task and instruction complexity effects. By definition, WM handles information processing, and cognitive load represents the *attention* required to handle information within WM, which determines the *easiness* of task completion. We identified two proxies corresponding to these key characteristics:

**Attention Entropy** measures the diversity of the model's attention distribution, with higher entropy indicating more distributed attention across input tokens, suggesting the model must consider multiple aspects of the input, corresponding to higher cognitive load (Zhang et al., 2025). **Perplexity** measures the model's certainty of solutions, serving as a proxy for the *easiness* of task completion. For the *Task Complexity Effect* experiment, we construct Q&A pairs from AMPS Hendrycks et al. (2021a) with 4 difficulty levels (simple to complex arithmetics), controlling input length for fair comparison. For the *Instruction Complexity Effect* experiment, we select Q&A pairs from FLASK Ye et al. (2023) with varying instruction complexity, measuring perplexity on answers for both hard and easy tasks. See Appendix B.2 for details of the experimental setup, results, and discussion of these proxies.

| Experiment 1: Task Complexity Effect. | | | Experiment 2: Instruction Complexity Effect. | | |
|---|---|---|---|---|---|
| **Task** | **Attention Entropy** | | **Instruction** | **PPL (Hard)** | **PPL (Easy)** |
| Level 1 | 4.44 | | Level 1 | 120.50 | 3.37 |
| Level 2 | 4.80 | | Level 2 | 88.97 | 3.42 |
| Level 3 | 5.04 | | Level 3 | 85.35 | 3.45 |

Table 1: Pilot study results: (Left) Attention entropy increases with task complexity, indicating higher cognitive load. (Right) Perplexity patterns are aligned with CLT predictions: instructions help reduce cognitive load for hard tasks but show no benefit for easy tasks.

The results provide strong empirical support for our CLT-LLM analogy. Attention entropy increases with task complexity (Table 1, left), indicating that harder tasks require the model to simultaneously consider more information pieces, corresponding to higher cognitive load. For perplexity (Table 1, right), hard tasks show decreasing perplexity with instruction complexity, indicating that instructions help the model focus and reduce cognitive load. Easy tasks show increasing perplexity, suggesting instructions provide no benefit and may even introduce extraneous load, validating CLT's redundancy effect where additional information impairs performance when task demands are within capacity. This aligns with CLT predictions about cognitive load management.

## 3.3 Collective Intelligence Principles for Cognitive Load Management

Having empirically validated the analogy, we can now leverage CLT principles to address cognitive overload in LLMs. As depicted in the final block of Fig. 1, when humans encounter tasks that exceed individual WM capacity, we employ two strategies: external tools or collective intelligence. For complex tasks where external tools are insufficient, humans naturally form collaborative cognitive systems that exceed individual capabilities, leading to the emergence of a *collective mind* that is more powerful than the sum of all individuals (Woolley et al., 2010; Malone et al., 2010). CLT provides principled guidance for managing cognitive load within such collective systems, particularly addressing how the introduction of new agents or coordination mechanisms can introduce extraneous load that must be carefully balanced. Appendix B.1.3 provides discussion from cognitive science.

This collective intelligence effectively manages cognitive load through three core mechanisms guided by CLT principles: (i) **Division of Cognitive Labor** through parallel thinking, allowing individuals to focus on specialized aspects of problems, thereby reducing intrinsic load per individual (Dunbar, 2003); (ii) **Collective Working Memory**, often through Transactive Memory Systems where knowledge and responsibilities are distributed, enabling individuals to rely on each other for information sharing while managing the extraneous load of coordination (Wegner, 1987; Kirschner

et al., 2018); and (iii) **Structured Communication** that efficiently integrates diverse insights through organized information flow, carefully managing extraneous load to prevent cognitive overload from coordination overhead (Hutchins, 1995). Since LLMs face analogous cognitive limitations and our pilot study demonstrates measurable cognitive load effects, systematically operationalizing these CLT principles in multi-agent LLM systems should analogously provide benefits.

## 4 COTHINKER

*CoThinker* is the operationalization of the CLT principles and collective intelligence mechanisms outlined in Section 3, designed as a multi-agent ICL architecture that systematically manages CL to enhance collaborative problem-solving. Simply aggregating outputs from LLM agents often proves insufficient for complex tasks, as naive collaboration can introduce transactional costs—the cognitive effort required to coordinate, communicate, and integrate—without a corresponding increase in solution quality (Pan et al., 2025). As CLT predicts, these transactional costs can quickly lead to cognitive overload, negating the benefits of parallel thinking (Kirschner et al., 2009; 2018). *CoThinker* addresses these challenges by translating the three core CLT principles into a practical multi-agent framework that creates a "collective mind" capable of distributing CL effectively.

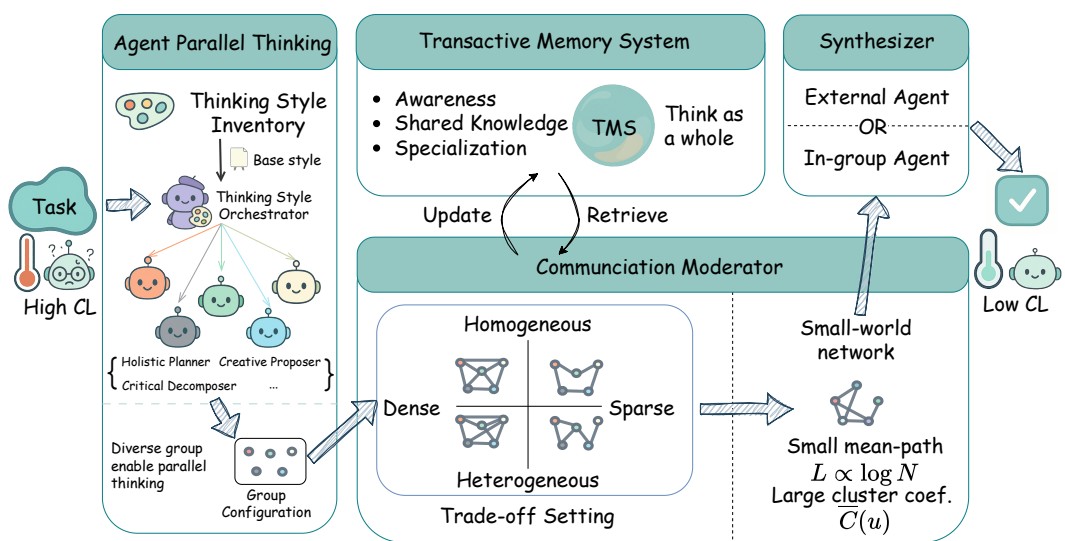

Figure 2: The *CoThinker* Architecture. A high CL task is initially processed by diverse agents via Agent Parallel Thinking. Transactive Memory System facilitates shared understanding by updating and retrieving of collective knowledge. Communication Moderator manages inter-agent information flow, leveraging a trade-off to form a cognitive small-world network, which then feeds into the Synthesizer for final solution, resulting in a lower effective CL for the overall system.

To operationalize these insights, the *CoThinker* architecture (Figure 2) comprises four main modules: Agent Parallel Thinking (Section 4.1), Transactive Memory System (Section 4.2), Communication Moderator (Section 4.3), and Synthesizer (Section 4.4). Each module is directly guided by CLT principles to emulate aspects of the human collective mind. Agent Parallel Thinking fosters initial cognitive diversity, potentially splitting the intrinsic load of the task. The Transactive Memory System boosts inter-agent understanding and tracks consensus, reducing CL from redundant processing. The Communication Moderator balances intrinsic and extraneous loads by structuring information exchange. Finally, the Synthesizer integrates refined collective insights. Let $\mathcal{A} = \{A_1, \ldots, A_M\}$ be the set of $M$ agents. Let $T_{\max}$ be the total number of generation rounds. Agent $A_i$'s output at the end of round $t$ is denoted $x_i^{(t)}$.

### 4.1 AGENT PARALLEL THINKING

This module promotes a *division of cognitive labor* and *parallel thinking* by assigning diverse thinking styles. Unlike assigning pre-defined roles, which require domain-specific foresight and

impose extraneous CL from role adherence, *CoThinker* uses an adaptive approach. A Thinking Style Orchestrator generates a task-specific style $\phi_i$ for each agent $A_i$ based on a general base thinking style inventory $\psi$ (Sternberg, 1997) and the task $D$:

$$\{\phi_i\}_{i=1}^M = \text{Orch}(D, \psi) \tag{1}$$

This yields diverse thinking styles $\{\phi_i\_i = 1^M$, employed in subsequent stages. Unlike pre-defined roles requiring complex persona maintenance (extraneous load), thinking styles (Sternberg, 1997) represent preferred ways of applying capabilities, enabling division of labor with minimal overhead. Further details on the prompting strategy for style generation and thinking style inventory are in Appendix D.3.

## 4.2 TRANSACTIVE MEMORY SYSTEM (TMS)

Human groups effectively manage complex information by developing Transactive Memory Systems (TMS), which involve a shared understanding of who knows what, how to access information held by others, and a collective agreement on the information itself (Wegner, 1987), hollingshead2001cognitive. This distributed cognitive system allows individuals to specialize and rely on others, reducing individual CL and enhancing group problem-solving (Lewis, 2003). To emulate these benefits and foster a *collective working memory* in *CoThinker*, we implement a structured mechanism for maintaining and accessing shared knowledge. This enables cognitive offloading—agents rely on shared "who knows what" knowledge (Wegner, 1987; Lewis, 2003)—and reduces coordination overhead by maintaining explicit expertise awareness. Implementation details in Appendix D.4. At each round $t$, an evolving representation of the group's collective knowledge, denoted $\mu^{(t)}$, is updated based on contributions from all agents:

$$\mu^{(t+1)} = \text{UpdateMem}(\mu^{(t)}, \{x_j^{(t)}\}_{j=1}^M) \tag{2}$$

## 4.3 COMMUNICATION MODERATOR

Effective inter-agent communication is crucial, yet it incurs transactional costs, the cognitive effort for message processing and integration, imposing extra extraneous CL. To mitigate these costs, Communication Moderator selects $N < M$ reference messages for each agent $A_i$. This process navigates the critical trade-offs between **Network Density vs. Sparsity** (high exposure vs. information loss) and **Information Homogeneity vs. Heterogeneity**. The latter involves balancing the ease of integrating cognitively similar inputs (low extraneous load but risk of echo chambers (Runkel, 1956)) against the benefits of diverse perspectives for distributing intrinsic load (Aral & Van Alstyne, 2011).

**Communication Topology and Algorithm:** The selection of references defines a directed communication graph $G^{(t-1)} = (\mathcal{A}, E^{(t-1)})$ for each round, where an edge $(A_u, A_v) \in E^{(t-1)}$ exists if agent $A_v$ receives a message from agent $A_u$ generated in round $t - 1$. Motivated by how small-world networks efficiently balance local clustering with global connectivity (Watts & Strogatz, 1998), our moderator employs the following algorithm to construct this graph:

a. **Fixed In-Degree** ($N$): Each agent $A_i$ (node $A_v$) has an in-degree of $N$, capping its processing load and respecting LLM WM (Zhang et al., 2024b; Gong et al., 2024).
b. **Define Cognitive Distance between Agent Outputs:** The cognitive distance $d(x_u^{(t-1)}, x_v^{(t-1)}) = 1 - \text{sim}(x_u^{(t-1)}, x_v^{(t-1)})$ is based on the semantic similarity of previous outputs.
c. **Re-connection via Probabilistic Rewiring** ($\beta$): For each agent $A_i$, its $N$ incoming edges (references $\mathcal{P}_i^{(t-1)}$) are chosen from cognitively similar peers (low distance), but with a probability $\beta$, "rewiring" some connections to randomly chosen, diverse peers.

**Resulting Network Properties and Cognitive Balance:** It fosters dynamic communication networks with small-world properties, with high local clustering (facilitating efficient refinement of similar ideas, reducing extraneous load locally) and short average path lengths (enabling rapid global propagation of diverse insights, aiding intrinsic load distribution). This structure offers a balance between focused collaboration and broad information access, managing CL more effectively than random or regular lattice networks. Further details are in Appendix D.5.

## 4.4 SYNTHESIZER

The Synthesizer consolidate all agents answer and TMS into a final answer (details in Appendix D.6). *CoThinker* **Process Flow.** The process for task $D$ with $M$ agents over $T$ rounds:

*Initialization:*

$$\{\phi_i\}_{i=1}^M = \text{Orch}(D, \psi_i), \quad x_i^{(0)} = \text{Agent}(D, \phi_i), \quad \mu^{(0)} = \text{UpdateMem}(\{x_i^{(0)}\}_{i=1}^M) \quad (3)$$

*Iterative Refinement for agent $A_i$ and round $t$:*

$$\mathcal{P}_i^{(t)} = \text{SelectRefs}(\{x_k^{(t)}\}, N, \beta) \quad (4)$$

$$x_i^{(t+1)} = \text{Agent}(D, \phi_i, \mu^{(t)}, x_i^{(t)}, \mathcal{P}_i^{(t)}), \quad \mu^{(t+1)} = \text{UpdateMem}(\mu^{(t)}, \{x_k^{(t+1)}\}) \quad (5)$$

*Final Synthesis:*

$$y_{\text{final}} = \text{Synth}(\{x_i^{(T-1)}\}_{i=1}^M, \mu^{(T-1)}, D) \quad (6)$$

## 5 EXPERIMENTS AND RESULTS

This section details our experimental methodology and presents the empirical evaluation of *CoThinker*. We first outline the experimental setup, and then present the main results on LiveBench (White et al., 2025) and CommonGen-Hard (Madaan et al., 2023), followed by ablation studies and a discussion of our findings through the lens of Cognitive Load Theory (CLT).

### 5.1 EXPERIMENTAL SETUP

**Models and Configuration.** For main experiments, we use three Gemini models (Team et al., 2024) with varying capacities: Gemini-1.5-Flash-8B (lightweight), Gemini-1.5-Flash (mid-tier), and Gemini-1.5-Pro (high-capacity). **Evaluation Benchmarks.** We evaluate on two challenging benchmarks: (1) *LiveBench* (White et al., 2025), a comprehensive SOTA benchmark containing *real-world tasks*, providing broad-spectrum evaluation (math, reasoning, data analysis and so on); and (2) *CommonGen-Hard* (Madaan et al., 2023), a controlled *experimental challenge* designed to test information interactivity, by forcing models to integrate target concepts from large pools of distractors. **Baselines.** We compare *CoThinker* with both single-agent and multi-agent approaches: Single Agent (IO), Single Agent (CoT) (Wei et al., 2022), Single Agent (Self-Refine) (Madaan et al., 2023), Multi-Agent Debate (MAD) (Du et al., 2023; Liang et al., 2023), and Diverse MAD (DMAD) (Liu et al., 2025b). Complete details are in Appendices E.3, and E.2.

### 5.2 MAIN RESULTS ON LIVEBENCH

Table 2 shows *CoThinker* performance across three Gemini models. Scores are normalized by the 8B model's IO baseline. *CoThinker* achieves strong average performance, excelling in complex tasks (Data Analysis, Reasoning, Math) but underperforming on Instruction Following. This pattern reflects distinct tasks with different cognitive load (CL) characteristics: (1) **High intrinsic CL tasks** show clear performance scaling with model capability increases, indicating cognitive bottlenecks, where *CoThinker* excels by distributing intrinsic CL across agents. (2) **Low intrinsic CL tasks** show minimal gains from stronger models. *CoThinker*'s communication overhead introduces extraneous CL that outweighs collaboration benefits, explaining underperformance on execution-focused tasks requiring straightforward adherence rather than reasoning.

### 5.3 MAIN RESULTS ON COMMONGEN-HARD

*CoThinker* demonstrates improvements on CommonGen-Hard, which tests high CL management. Figure 3 shows performance across evaluation dimensions through (a) a radar plot with normalized by dividing the min scores and (b) an interaction rounds plot tracking performance evolution. *CoThinker* effectively handles high element interactivity by distributing CL across specialized agents and leveraging transactive memory—core CLT principles in action. The radar plot (Figure 3a) reveals *CoThinker*'s strengths in coherence and concept integration, with minor trade-offs in conciseness.

| Task | Gemini-1.5-Flash-8B | | | | | | Gemini-1.5-Flash | | | | | | Gemini-1.5-Pro | | | | | |
|---|---|---|---|---|---|---|---|---|---|---|---|---|---|---|---|---|---|---|
| | $IO$ | $CoT$ | $SR$ | $MAD$ | $DMAD$ | $Ours$ | $IO$ | $CoT$ | $SR$ | $MAD$ | $DMAD$ | $Ours$ | $IO$ | $CoT$ | $SR$ | $MAD$ | $DMAD$ | $Ours$ |
| Math | 1.00 | 1.04 | 0.92 | **1.13** | **1.13** | 1.11 | 1.47 | 1.47 | 1.45 | 1.51 | 1.49 | **1.57** | 2.00 | 1.86 | 1.93 | 2.29 | 2.31 | **2.40** |
| Data | 1.00 | 0.90 | 0.34 | 0.58 | 0.64 | **1.32** | 2.03 | 2.07 | 0.90 | 1.46 | **2.51** | 2.44 | 2.92 | 2.72 | 1.33 | 3.15 | 3.32 | **3.39** |
| Reas. | 1.00 | 1.11 | 0.80 | 1.21 | 0.85 | **1.22** | 1.63 | 1.74 | 1.55 | 1.92 | 1.94 | **1.97** | 1.87 | 1.82 | 1.80 | 1.78 | 1.88 | **1.95** |
| Lang. | 1.00 | **1.09** | 0.89 | 1.03 | 1.02 | 0.98 | 1.41 | 1.30 | 1.06 | 1.46 | 1.44 | **1.52** | 1.43 | 1.54 | 1.22 | 1.58 | 1.74 | **1.76** |
| Instr. | 1.00 | **1.02** | 0.81 | 0.87 | 0.89 | 0.80 | **1.10** | **1.10** | 0.87 | 1.01 | 1.06 | 0.99 | **1.03** | 1.02 | 0.72 | 0.77 | 1.02 | 0.95 |
| Avg. | 1.00 | 1.03 | 0.75 | 0.97 | 0.91 | **1.07** | 1.53 | 1.54 | 1.17 | 1.47 | 1.69 | **1.70** | 1.85 | 1.79 | 1.40 | 1.92 | 2.05 | **2.09** |

Table 2: LiveBench (White et al., 2025) performance normalized by Gemini-8B (IO). *Ours* refers to CoThinker. Abbreviations: Math, Data Analysis, Reasoning, Language, and Instruction Following.

The rounds plot (Figure 3b) shows sustained improvement across multiple interaction rounds: while baseline methods degrade due to accumulated coordination overhead (extraneous CL).

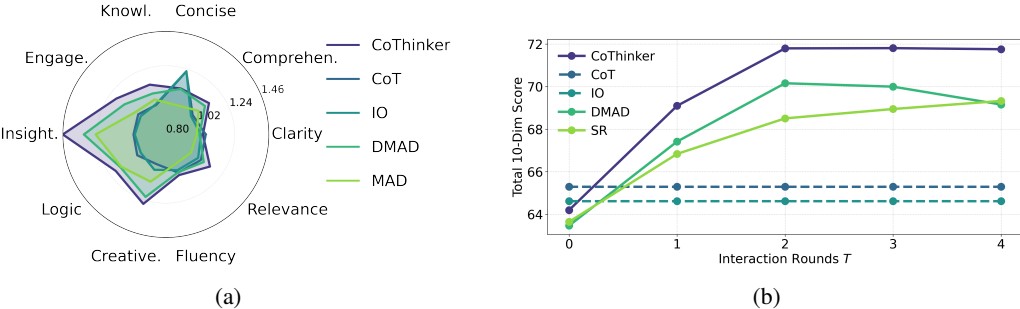

(a) (b)

Figure 3: *CoThinker* performance on CommonGen-Hard (Madaan et al., 2023) using Gemini-1.5-Flash. (a) The radar plot illustrates a multi-dimensional performance, showing well-rounded improvement. (b) The rounds plot depicts the total score across rounds, showing stable improvement.

## 5.4 Cross-Model Generalization

To validate *CoThinker*'s model-agnostic nature, we evaluated across multiple LLM families: GPT-5-Nano, Qwen3-30B-A3B, GPT-OSS-20B, Gemini-2.5-Flash, GPT-4.1-Mini, Qwen3-32B, and DeepSeek-R1-8B (partial results in Table 3). We examine two scenarios: (1) standard setting with IO baselines using maximum reasoning steps with temperature 0.25 and (2) constrained setting with token budget of 8192 with greedy decoding (temperature = 0). Complete results are in Appendix E.4.

| | Standard Setting | | | | | Constrained Setting | | |
|---|---|---|---|---|---|---|---|---|
| **Model** | **Method** | **Math** | **Reason.** | | **Model** | **Method** | **Math** | **Reason.** |
| GPT-5 | CoThinker | **88.57** | **81.88** | | Gemini-2.5 | CoThinker | **76.3** | **69.2** |
| | IO | 82.63 | 68.38 | | | IO | 59.3 | 31.0 |
| Qwen3 | CoThinker | **80.62** | **89.50** | | GPT-4.1 | CoThinker | **40.0** | **70.8** |
| | IO | 77.50 | 76.00 | | | IO | 34.0 | 40.8 |

Table 3: Cross-model evaluation on LiveBench (White et al., 2025) Math and Reasoning (Reason.) subsets in standard setting (left) and constrained setting (right).

## 5.5 CoThinker Ablation of Communication Moderator

We ablate the Communication Moderator's key parameters—reference set size ($N$), exploration rate ($\beta$), and agent count ($M$)—on Gemini-1.5-Flash-8B across four LiveBench categories: Math,

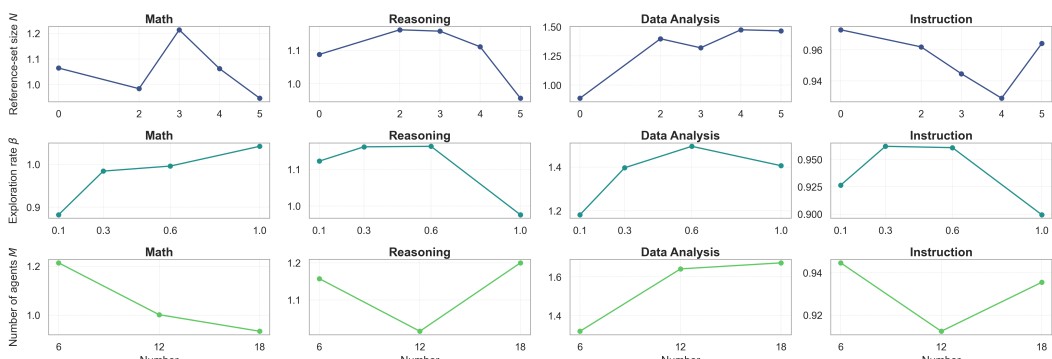

Figure 4: Communication Moderator ablation on *CoThinker* using Gemini-1.5-Flash-8B. **Top**: Reference Set Size ($N \in \{0, 2, 3, 4, 5\}$, $M = 6, \beta = 0.3$); **Middle**: Exploration Rate ($\beta \in \{0.1, 0.3, 0.6, 1.0\}$, $N = 2, M = 6$); **Bottom**: Agent Count ($M \in \{6, 12, 18\}$, $N = 3, \beta = 0.3$).

Reasoning, Data Analysis, and Instruction (See Figure 4). Default parameters: $T = 3$, with controlled variations for each parameter. Scores are normalized by IO baseline. **Analysis.** The Communication Moderator's parameters directly control CL balance: *Reference set size ($N$)* manages extraneous load—optimal $N = 2-3$ balances peer input diversity against overload, respecting LLM working memory limits. *Exploration rate ($\beta$)* governs similarity-diversity trade-offs: low $\beta$ (exploiting similar ideas) reduces integration load but risks echo chambers; high $\beta$ (exploring diverse perspectives) aids intrinsic load distribution but increases extraneous load. Task-dependent optima (e.g., higher $\beta$ for Reasoning) reflect this balance through small-world network properties. *Agent count ($M$)* shows non-monotonic performance—more agents distribute intrinsic load but elevate coordination costs, confirming CLT predictions for group overload. These findings validate the Communication Moderator's role in managing CL for effective collective intelligence. See Appendix E.6 for details.

## 5.6 COTHINKER ABLATION ON OTHER COMPONENTS

We conducted component ablation on Transactive Memory System (TMS) and Thinking Style Orchestrator (Style) a subset of LiveBench Math tasks. We also examine the proxy, perplexity (PPL), to reflect the CL management effects of our components.

**Component Ablation.** Communication Moderator is fixed ON for all runs. We test configurations: *TMS* $\in \{$On, Off$\}$; *Style* $\in \{$On, Off$\}$. **Analysis.** Our components benefit most models (details in Appendix E.5). From Table 4, we find Thinking Style Orchestrator provides consistent improvements. However, TMS is less effective for new GPT models. Investigating their output, we find that they often refuse to give intermediate results, responding with "I can't provide step-by-step reasoning." This is counterproductive, as it discourages the detailed reasoning used to build TMS.

| Configuration | Qwen3-30B-A3B | GPT5-Nano | GPT-OSS-20B |
|---|---|---|---|
| TMS: ON, Styles: ON | 81.87 | 55.97 | 57.15 |
| TMS: ON, Styles: OFF | 69.79 | 49.02 | 48.21 |
| TMS: OFF, Styles: ON | 76.41 | 62.36 | 58.37 |

Table 4: Component ablation on subset of Math dataset, with effect of TMS and Thinking Style Orchestrator, indicating the CL management benefits of each component.

**PPL proxy as Evidence..** We conducted perplexity (PPL) studies on weaker models to demonstrate how our components helps weaker models reduce CL in understanding stronger models' outputs. We choose Math and Reasoning tasks for our analysis. Lower PPL indicates higher easiness and effective CL reduction (See Table 5). For more interesting ablation with PPL proxy, see Appendix E.5.

| Model | Baseline | Styles | TMS | References (N=3) |
|---|---|---|---|---|
| Qwen3-8B | 6.56 | 3.58 | 1.69 | 3.10 |
| Mistral-7B | 6.63 | 5.04 | 1.58 | 2.86 |

Table 5: PPL ablation showing CL reduction effects: how components help weaker models process information from better models' answers, reducing CL (lower PPL).

## 6 CONCLUSION

This work explains ICL performance ceilings in LLMs through Cognitive Load Theory (CLT), attributing failures on complex tasks to cognitive overload. We establish both theoretical foundations mapping human working memory to LLM attention mechanisms and empirical examination through measurable cognitive load proxies (attention entropy and perplexity), demonstrating that the CLT-LLM analogy provides a principled framework for understanding and addressing performance limitations. We introduce *CoThinker, a CLT-grounded multi-agent architecture with specialized agents, a transactive memory, and communication moderator to reduce load. On benchmarks with complex tasks, CoThinker consistently outperforms strong baselines. Analysis confirm effective cognitive load management, suggesting a principled CLT-based path to more capable collaborative LLM systems. Our work complements recent multi-agent advances (Wang et al., 2023; Suzgun & Kalai, 2024) by providing theoretical grounding in CLT, generating falsifiable predictions and deriving principled design constraints rather than relying on heuristics.*

## ETHICS STATEMENT

This research contributes to society and human well-being by advancing scientific understanding of cognitive load principles in AI systems, potentially enabling more capable and efficient collaborative AI. We uphold high standards of scientific excellence through rigorous experimental design, transparent reporting of methods and results, and honest acknowledgment of limitations. The work avoids harm by focusing on computational improvements without involving human subjects, sensitive personal data, or applications with direct societal risks. We maintain honesty and transparency by providing complete implementation details, acknowledging all limitations, and declaring no conflicts of interest. The research promotes fairness by using publicly available benchmarks and established evaluation protocols accessible to the broader research community. We respect intellectual property by properly citing all prior work and acknowledging the foundation provided by existing research. Privacy is honored as no personal data collection or processing occurs. All experimental data uses publicly available benchmarks and API-based LLM services under their respective terms of service.

## REPRODUCIBILITY STATEMENT

Our methods rely primarily on existing LLM APIs where we have detailed the specific models used and provided sample prompts and exact agent workflow descriptions in the main paper and appendix. Complete implementation details for reproducing our results are provided, including all agent prompts, communication protocols, system configurations, and evaluation procedures. The benchmark datasets (LiveBench, CommonGen-Hard) are publicly available with standardized evaluation metrics that enable direct comparison.

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

# Appendix

## A  LLM USAGE

LLMs served as experimental subjects for evaluating our cognitive load theory hypotheses (not as sources of research ideation). They were additionally used in limited code authoring support and writing assistance (grammar correction and standard-English rephrasing). They were not used for generating core methodological ideas, designing experiments, forming hypotheses, performing analysis, or implementing core architectural components. All conceptual framing, system design, analysis, and conclusions are entirely human-generated. Authors retain full responsibility for all content.

## B  THEORETICAL FOUNDATIONS AND VALIDATION

### B.1  COGNITIVE FOUNDATIONS AND THE CLT-LLM ANALOGY

Here we elaborate on the cognitive science foundations underpinning our framework and the analogy to LLMs from Section 3.1.

#### B.1.1  HUMAN WORKING MEMORY AND ATTENTIONAL CONTROL

Human working memory (WM) is a core cognitive faculty for actively holding and manipulating a limited amount of information relevant to ongoing tasks, operating through attentional mechanisms that select and maintain internal representations, often associated with sustained neural activity in regions like the prefrontal cortex (Baddeley et al., 1986a; Cowan, 2010; Postle, 2006). Given that Large Language Models exhibit emergent sparse attention—where specific attention heads specialize in processing distinct patterns rather than diffusely attending to all input tokens (Vaswani et al., 2017; Voita et al., 2019)—it prompts an intriguing question: does this selective information processing within a finite context window imply the existence of a functional analogue to human WM in LLMs? This emergent selectivity, where not all information in the context is equally weighted or actively processed at any given step, forms a crucial part of the analogy we draw to understand potential capacity limitations and cognitive load phenomena in these models, particularly when handling tasks with high element interactivity through In-Context Learning.

#### B.1.2  WORKING MEMORY AND PARAMETRIC KNOWLEDGE IN LLMS

**Misunderstanding 1: Working Memory as Context Window Limits** A prevalent misunderstanding in current LLM research equates Working Memory (WM) with physical limitations like context window size and assumes that models only fail when they exceed their maximum token limits—that WM is simply the amount of in-context information that can fit within the context window. However, our framework distinguishes WM as the mechanism of selective attention: a limited cognitive capacity to actively hold and manipulate information simultaneously, not merely passive storage of tokens. This selective attention mechanism can become overwhelmed and lead to performance degradation well before reaching physical context limits.

**Misunderstanding 2: High Cognitive Load as Long Input/Output Tasks** Another common misconception is that high cognitive load (CL) tasks are simply those with long inputs and long outputs. This conflates task length with cognitive complexity. In our framework, a high CL task is defined by the working memory demands it places on simultaneous information processing—specifically, the amount of information that must be actively held and manipulated *simultaneously* in WM. For example, a sequential fact elicitation task (e.g., "List 100 countries and their capitals") may have long input and output but requires minimal WM since each fact can be retrieved independently without integrating interdependent information. Conversely, a complex reasoning task requiring simultaneous consideration of multiple interacting constraints, relationships, or variables creates high CL regardless of input/output length, as it demands that the model maintain and manipulate multiple information pieces concurrently in its working memory.

**Our Framework: Working Memory and Parametric Knowledge** Our framework clarifies that an LLM's parameters can be understood as 1)*Parametric Knowledge* (analogous to human long-term

memory, mainly from FFN layer Geva et al. (2021)), and 2) innate generation capabilities that determine how effectively the model can select and process information for its WM (mainly attention layer Tighidet et al. (2025)). A model's parametric knowledge affects how well it understands and integrates information during reasoning (reflected in attention patterns), while WM serves as the active processing resource that must load both contextual information and activated parametric knowledge Tighidet et al. (2025). Complex tasks demanding simultaneous consideration of multiple interacting elements can cause cognitive overload regardless of context window utilization or task length. Our pilot study (Appendix B.2) provides empirical validation of this WM framework through measurable cognitive load proxies.

### B.1.3 Detailed Mechanisms of Human Collective Intelligence

Here we elaborate on the cognitive mechanisms enabling human collective intelligence from Section 3.1. Human collective intelligence emerges from sophisticated social-cognitive abilities that enable groups to surpass individual cognitive limitations through several key mechanisms:

**Shared Intentionality and Theory of Mind:** Effective collective intelligence requires individuals to understand others' mental states and coordinate intentions toward common goals (Tomasello et al., 2005; Frith & Frith, 2005), enabling the establishment of common ground necessary for distributed cognitive processing.

**Meta-cognitive Awareness:** Individuals develop meta-knowledge about "who knows what" (Hollingshead, 2001), enabling efficient allocation of cognitive resources and allowing group members to rely on each other for information sharing and retrieval (Hollingshead & Brandon, 2003).

**Spontaneous Organization:** Effective collective intelligence often emerges spontaneously through self-organizing principles (Shteynberg et al., 2023), with groups naturally developing communication patterns and role distributions that optimize cognitive load management.

**Structured Communication:** Groups develop specialized communication protocols that establish common ground, minimizing extraneous cognitive load while maximizing information integration (Tomasello et al., 2005).

These mechanisms demonstrate that human collective intelligence results from emergent properties of social-cognitive interaction that specifically address cognitive load management challenges, providing natural solutions to cognitive overload that inform LLM multi-agent system design.

### B.1.4 Using Cognitive Load Theory to Explain Phenomena in LLM Performance

Here we provide examples of how CLT explains LLM performance phenomena from Section 3.1. Cognitive Load Theory (CLT) offers a valuable lens to interpret puzzling LLM performance issues, positing that LLMs, like humans, have finite processing capacity. Exceeding this capacity leads to performance degradation. This section concisely analyzes several such cases through CLT.

1. **Degradation of Thought in Self-Reflection:** Liang et al. (2023) found LLMs may rigidly stick to incorrect initial answers during self-reflection, failing to correct meaningfully.
   - *CLT Explanation:* Self-reflection (holding problem, solution, critique, and revision process concurrently) is highly demanding. If initial analysis already consumes most capacity, the LLM may lack resources for genuine re-evaluation, defaulting to superficial agreement due to cognitive overload.
2. **Performance Degradation with More In-Context Examples (Many-Shot ICL):** Agarwal et al. (2024) noted LLM performance can degrade with more in-context examples, especially on complex tasks (e.g., MATH).
   - *CLT Explanation:* While few examples scaffold, excessive examples increase total cognitive load beyond capacity. The LLM struggles to synthesize all information, akin to CLT's "redundancy effect" where too much information, even relevant, overwhelms working memory.
3. **Performance Degradation Despite Increasing "Confidence" (NLL Trends):** Agarwal et al. (2024) also found that performance degradation in many-shot ICL wasn't always explained by NLL (confidence) trends; NLL could improve as performance worsened.
   - *CLT Explanation:* Under cognitive overload, LLMs (like humans) may resort to heuristics. Overwhelmed by many examples, an LLM might latch onto superficial patterns, yielding outputs

that are stylistically plausible (good NLL) but incorrect. This "overconfidence" in a flawed heuristic stems from an inability to allocate resources for deeper reasoning.

4. **Reduced Diversity after RLHF for Instruction Following:** Kirk et al. (2023) and others observed that RLHF, while improving instruction following, can reduce output diversity.
   - *CLT Explanation:* Intense RLHF training on narrow preferences imposes a high "germane load" for conformance. To manage this, and the extraneous load of deviating from rewarded paths, the model may operate in a constrained output space, reducing the cognitive effort of exploring diverse (potentially unrewarded) responses. The "cost" of diversity becomes too high.

These instances suggest CLT is a powerful analogical framework for understanding LLM limitations under demanding informational or processing conditions.

## B.2 PILOT STUDY: EMPIRICAL VALIDATION

This section provides detailed implementation and analysis of our pilot study that empirically validates the Cognitive Load Theory (CLT) analogy for LLMs, as summarized in Section 3.2.

## B.3 MATHEMATICAL DEFINITIONS OF COGNITIVE LOAD PROXIES

**Attention Entropy** measures the diversity of the model's attention distribution, formally defined as:

$$H = -\sum_{i=1}^{N} a_i \log a_i \tag{7}$$

where $a_i$ is the normalized attention weight on token $i$ and $N$ is the total number of tokens. Higher entropy indicates more uniform attention distribution across input tokens, suggesting the model must consider more aspects of the input simultaneously.

**Perplexity** measures the model's uncertainty about its predictions, formally defined as:

$$\text{PPL} = \exp\left(-\frac{1}{N}\sum_{i=1}^{N} \log P(w_i)\right) \tag{8}$$

where $P(w_i)$ is the probability assigned to token $w_i$ and $N$ is the sequence length. Lower perplexity indicates higher confidence in predictions.

## B.4 THEORETICAL JUSTIFICATION FOR PROXIES

The two proxies we employ measure **LLM-perceived cognitive load** that must be accommodated within the model's working memory for task completion. This distinction is crucial for understanding what we are measuring. Our analogy starts with the observation that the human brain can only attend to a limited amount of information at once, a feature known as working memory. In cognitive science, working memory capacity refers to how many elements one can simultaneously hold and manipulate; these elements are often referred to as "information chunks." A high cognitive load task requires more working memory as it demands the simultaneous use of more elements to solve. Therefore, the CL of a task can be reflected by how many "information chunks" are needed to solve it.

**Why Attention Entropy? (Measuring "Information Chunks"):** The architecture of LLMs, due to their inherent attention mechanism, similarly restricts the amount of information they can focus on at once. LLMs predict the next tokens through their attention mechanism, allowing the model to selectively focus on specific, relevant parts of the input context to generate a response. By definition, we can measure how many "information chunks" are being actively considered to complete a task. Thus, using Attention Entropy to measure the sparsity of information integration is a direct proxy:

- Low Entropy = Sparse/Focused attention = The model only needs few "elements" to predict the answer (Low Load).
- High Entropy = Uniform attention = The model is attending to many distinct "elements" simultaneously to predict the answer (High Load).

This analogy is justified through both our experiments and recent theoretical work. In our experiments, we find attention entropy correlates with task complexity (Table 6); also we find adding reasoning

effectively reduces attention entropy even with longer context. Our analogy also matches recent work in theoretically explaining why chain of thoughts works (Wen et al., 2025a); they also find CoT is creating more sparse attention; in terms of our framework, it is that reasoning is creating cognitive offloading, allowing the model to process fewer chunks during solving the tasks.

**Why Perplexity? (Measuring Processing Fluency):** Perplexity serves as a proxy for processing fluency, representing the cognitive ease of processing information. Metacognitive research establishes that processing fluency (confidence) correlates with task difficulty and can reflect cognitive load faithfully when the subject is not under cognitive overload (Koriat, 2007). In our pilot study, we specifically measure perplexity on prefilled ground-truth answers and validate it. In contrast, during open generation, cognitive overload causes models to lose metacognitive calibration, often resulting in low-perplexity hallucinations where confidence fails to predict performance (An et al., 2024). Both LLM phenomena align with cognitive science findings on metacognitive judgments under varying load conditions.

## B.5   EXPERIMENTAL SETUP

**Model:** We conducted experiments using Mistral-7B-v0.3, a mid-sized language model that exhibits clear cognitive limitations while maintaining reasonable performance across diverse tasks.

**Prefilled vs. Generated Answers:** We measure perplexity on prefilled ground-truth answers rather than model-generated responses to avoid heuristic confounds: cognitively overloaded models may produce confident but incorrect responses (Agarwal et al., 2024), which would not reflect the true cognitive load of the underlying task.

**Implementation Details:** For Attention Entropy, we aggregate softmax attention weights, then average over heads and layers.

## B.6   EXPERIMENT 1: ATTENTION ENTROPY AND TASK COMPLEXITY

**Dataset Construction:** We constructed a controlled dataset from AMPS-Hard, focusing on arithmetic reasoning tasks. To ensure fair comparison, we:

- Selected 4 difficulty levels (simple arithmetic to complex multi-step problems)
- Controlled input length across difficulty levels to isolate complexity effects
- Ensured consistent question format and domain (mathematical reasoning)

Different question types are not directly comparable using attention entropy, as the metric is sensitive to input structure and content. Therefore, we focused on a single domain with graduated difficulty.

| Task Complexity | Attention Entropy (No Reasoning) | Attention Entropy (With Reasoning) |
|---|---|---|
| Level 1 | 4.442 | 4.439 |
| Level 2 | 4.796 | 4.726 |
| Level 3 | 5.043 | 4.937 |
| Level 4 | 6.101 | 5.920 |

Table 6: Attention entropy increases with task complexity, validating the proxy. Reasoning steps reduce entropy by helping the model focus on key information.

**Analysis.** Attention entropy increases monotonically with task complexity, indicating that harder tasks require the model to consider more information pieces simultaneously, corresponding to higher cognitive load.

Importantly, this finding is **mathematically non-trivial**. Standard information theory predicts that for a probability distribution over $N$ tokens, attention entropy $H = -\sum a_i \log a_i$ should increase approximately as $\log N$ when $N$ grows large. Thus, longer sequences naturally yield higher entropy. However, Table 6 reveals that adding reasoning steps actually *decreases* attention entropy (e.g., Level 3: 5.043→4.937) despite creating longer context. This seemingly paradoxical result demonstrates that reasoning induces *sparse attention patterns*—the model learns to chunk information into coherent reasoning steps, allowing it to focus on fewer elements simultaneously even as total context grows.

This finding aligns with recent theoretical work showing Chain-of-Thought creates sparse sequential dependencies that enhance processing efficiency, and validates our working memory analogy: just as humans chunk information to expand effective WM capacity, LLMs achieve similar cognitive offloading through structured reasoning.

### B.7 EXPERIMENT 2: PERPLEXITY AND INSTRUCTION COMPLEXITY

**Dataset Construction:** We used the FLASK dataset, specifically filtering for problems within the same category to ensure comparability. We selected pairs of problems where one was marked as requiring "expert knowledge" and another without this marking, representing hard and easy tasks respectively within the same domain.

**Instruction Complexity Levels:**

- Level 1: Blank (no additional instruction)
- Level 2: "Think step by step"
- Level 3: "Please think step by step, focusing on factuality and logical reasoning"
- Level 4: Extended reasoning guidance with specific cognitive strategies
- Level 5: Comprehensive instruction with multiple reasoning frameworks

**Analysis.** For hard tasks, perplexity initially decreases with instruction complexity (Levels 1-3), indicating that structured guidance helps the model focus and reduces cognitive uncertainty. However, perplexity increases again at higher instruction levels (Levels 4-5), suggesting that excessive instruction becomes an additional cognitive burden—classic extraneous load as predicted by CLT.

For easy tasks, perplexity remains consistently low and slightly increases with instruction complexity, indicating that additional guidance provides no benefit and may even introduce unnecessary extraneous load. This pattern directly validates CLT's **"redundancy effect"**: when a task is within the model's working memory capacity, additional information—even if relevant—can impair performance by consuming limited cognitive resources without providing commensurate benefits. The differentiated effect across difficulty levels is particularly telling. For hard tasks, instructions reduce perplexity from 120.50 to 85.35, indicating they help the model focus and reduce cognitive uncertainty. For easy tasks, the slight increase ($3.37 \rightarrow 3.45$) suggests that processing unnecessary guidance creates extraneous load that outweighs any potential benefit when the model's existing capacity already suffices.

| Instruction Complexity | Perplexity (Hard) | Perplexity (Easy) |
|---|---|---|
| Level 1 | 120.50 | 3.37 |
| Level 2 | 88.97 | 3.42 |
| Level 3 | 85.35 | 3.45 |
| Level 4 | 92.48 | 3.46 |
| Level 5 | 100.71 | 3.46 |

Table 7: Perplexity patterns validate CLT predictions across instruction complexity levels.

### B.8 PROXY SCOPE: DIAGNOSTIC, NOT OPERATIONAL

**Why not test-time use.** The proxies we study (attention entropy and perplexity) are not universally applicable at inference time. First, when cognitively overloaded, models may fall back on *heuristics*, yielding confident but incorrect responses (Agarwal et al., 2024); this breaks the assumption that a model's own outputs reveal the true load of the underlying task. Second, perplexity requires ground-truth answers, which are unavailable during inference. Together, these factors also clarify why using PPL as a test-time assistant tends to fail: heuristic-driven outputs can appear easy for the model while still being wrong, and ground-truth-based PPL cannot be computed online.

**How we ensured clean experimental signals.** In our experiments, we treat these proxies strictly as *diagnostic* rather than operational signals. To mitigate confounds, we (i) controlled task domain and type within each study (addressing cross-task incomparability), (ii) controlled input length within each complexity level (addressing length effects on both metrics), and (iii) evaluated on prefilled

ground-truth answers rather than model-generated outputs to avoid heuristic confounds. Consequently, the proxies serve to validate the theoretical foundations, not to drive the system at test time—hence *CoThinker* emphasizes architectural solutions over real-time cognitive load monitoring.

### B.9 POST-HOC ANALYSIS ON MAIN EXPERIMENTS USING PROXIES

We leverage perplexity as a cognitive load proxy to understand how different CoThinker components help weaker models process information from stronger models' outputs. This analysis provides direct evidence that our components reduce cognitive burden during collaboration.

This section (Table 8, Table 10, and Table 9) provides the complete set of PPL ablation studies for component analysis referenced in Section 5.6.

**Analysis.**

- **Component effects (TMS and Style).** Table 8 shows that both Thinking Style Orchestrator and TMS reduce PPL, indicating that each component lowers the cognitive effort required for the model to process peer information. Consistent with our captions, TMS yields the largest PPL relief, reflecting stronger scaffolding for integrating specialized knowledge.
- **Number of references ($k$).** Table 9 shows PPL decreases as the number of peer answers increases, with diminishing returns beyond moderate $k$. This suggests that models find it easier to understand additional similar information, but overly large $k$ risks introducing unnecessary content reducing effective using of these answers, as explained in Section 4.3.
- **Selection strategy at $k=3$.** Table 10 shows that selecting *similar* peers achieves the lowest PPL, random also helps, while fully *diverse* peers increase PPL due to higher integration cost. As discussed in Section 4.3 (and Appendix D.5), prioritizing similarity reduces extraneous load but risks echo chambers; probabilistic rewiring ($\beta$) adds diversity to distribute intrinsic load while preserving small-world efficiency.

| Model | Base PPL | +Style PPL | +TMS PPL |
|---|---|---|---|
| Mistral-7B | 6.6314 | 5.0422 | 1.5811 |
| Qwen3-8B | 6.5633 | 3.5814 | 1.6876 |

Table 8: PPL changes from baseline when adding Style or TMS. Both components reduce PPL; TMS shows the largest relief.

| $k$ **peers** | **Mistral-7B PPL** | **Qwen3-8B PPL** |
|---|---|---|
| 1 | 4.1516 | 3.3174 |
| 2 | 3.1343 | 3.1022 |
| 3 | 2.8578 | 3.0995 |
| 4 | 2.6123 | 2.5367 |
| 5 | 2.1221 | 2.1429 |
| 6 | 2.0917 | 1.8572 |

Table 9: PPL vs number of peer answers ($k$). Increasing $k$ consistently lowers PPL with diminishing returns beyond $k \geq 5$.

| Selection ($k=3$) | Mistral-7B PPL | Qwen3-8B PPL |
|---|---|---|
| Random (3rand) | 2.0967 | 2.1924 |
| Similar (3sim) | 1.8891 | 1.5399 |
| Diverse (3diverse) | 3.0958 | 2.7905 |

Table 10: Effect of selection strategy at $k=3$. Similar peers yield the lowest PPL; random also helps; fully diverse peers show higher PPL, consistent with higher integration cost.

# C  SMALL-WORLD NETWORK PROPERTIES AND STATISTICAL RIGOR

## C.1  SMALL-WORLD NETWORK ANALYSIS

To validate that our Communication Moderator creates small-world network properties as predicted by CLT, we analyze the emergent communication networks using the small-world coefficient $\sigma = (C/C_{rand})/(L/L_{rand})$ (Humphries & Gurney, 2008), where $C$ is clustering coefficient, $L$ is average path length, and the denominators are random graph baselines.

At each round $t$, we construct a directed weighted graph $G^{(t)} = (\mathcal{A}, E^{(t)}, W^{(t)})$ where edge weights $w_{uv} = 1 - \text{sim}(x_u, x_v)$ represent cognitive distance. We calculate clustering coefficient, path length, and compare against 100 random graphs with the same degree distribution.

All configurations yield $\sigma > 1$, confirming small-world properties. For M=6, N=3, $\beta$=0.3: median $\sigma$=2.75; M=12: $\sigma$=2.87; M=18: $\sigma$=3.12. This validates that the Communication Moderator successfully creates high local clustering (efficient refinement of similar ideas) with short average paths (rapid global propagation of diverse insights), as predicted by CLT.

## C.2  STATISTICAL SIGNIFICANCE ANALYSIS

**Bootstrap Standard Error Methodology.** We use bootstrap resampling to estimate the standard error (SE) of our performance metrics across test instances. We use greedy decoding (temperature=0) for all refinement rounds to ensure deterministic behavior. Initial generation uses temperature=0.25 to create diverse agent starting points, but subsequent refinement is deterministic. Multiple random seeds would yield highly similar results under greedy decoding. Therefore, we report bootstrap variance representing instance-wise variance under large samples rather than seed variance.

For each task and model, we: (1) collect raw scores across test instances, (2) perform bootstrap resampling with replacement (n=1000 iterations), (3) compute mean performance for each bootstrap sample, (4) calculate SE as standard deviation of bootstrap distribution, and (5) construct 95% confidence intervals using the percentile method.

**Bootstrap Standard Error Results.**

Table 11 shows complete bootstrap SE results for our main comparison. The small sampling variance across all methods and model families demonstrates high reliability and stability of our bootstrap estimates.

**Statistical Significance on High Cognitive Load Tasks.**

We computed 95% confidence intervals for key comparisons on high-CL tasks (Reasoning, Math). Non-overlapping confidence intervals indicate statistical significance at p<0.01. Statistical significance analysis shows:

- **Gemini-2.5-Flash on Reasoning:** CoThinker 75.6 (SE=4.15) vs IO 41.5 (SE=4.86). 95% CIs: [67.5, 83.7] vs [31.9, 51.1]. Non-overlapping CIs confirm p<0.01.
- **GPT-4.1-Mini on Reasoning:** CoThinker 76.4 (SE=3.90) vs IO 47.4 (SE=4.82). 95% CIs: [68.8, 84.0] vs [37.9, 56.9]. Non-overlapping CIs confirm p<0.01.
- **Qwen3-32B on Reasoning:** CoThinker 30.6 (SE=4.22) vs IO 11.0 (SE=3.14). 95% CIs: [22.3, 38.9] vs [4.8, 17.2]. Non-overlapping CIs confirm p<0.01.
- **DeepSeek-R1-8B on Reasoning:** CoThinker 20.4 (SE=3.58) vs IO 5.4 (SE=2.00). 95% CIs: [13.4, 27.4] vs [1.5, 9.3]. Non-overlapping CIs confirm p<0.01.

All comparisons on high-cognitive-load tasks demonstrate statistically significant improvements, validating CoThinker's effectiveness at managing cognitive load.

| Model | Method | Math | Data | Reasoning | Language | Inst. Follow. | Overall |
|---|---|---|---|---|---|---|---|
| Gemini-Flash-8B | IO | 0.139 | 0.146 | 0.180 | 0.177 | 0.038 | 0.043 |
| | CoT | 0.146 | 0.149 | 0.185 | 0.178 | 0.037 | 0.042 |
| | SR | 0.145 | 0.116 | 0.164 | 0.171 | 0.043 | 0.041 |
| | MAD | 0.144 | 0.146 | 0.186 | 0.152 | 0.043 | 0.043 |
| | DMAD | 0.138 | 0.140 | 0.189 | 0.174 | 0.043 | 0.042 |
| | CoThinker | 0.156 | 0.140 | 0.209 | 0.288 | 0.040 | 0.045 |
| Gemini-Flash | IO | 0.172 | 0.148 | 0.209 | 0.213 | 0.036 | 0.044 |
| | CoT | 0.167 | 0.143 | 0.211 | 0.193 | 0.036 | 0.045 |
| | SR | 0.158 | 0.103 | 0.204 | 0.207 | 0.043 | 0.045 |
| | MAD | 0.165 | 0.136 | 0.215 | 0.204 | 0.041 | 0.045 |
| | DMAD | 0.169 | 0.141 | 0.216 | 0.204 | 0.037 | 0.044 |
| | CoThinker | 0.191 | 0.147 | 0.231 | 0.311 | 0.038 | 0.045 |
| Gemini-Pro | IO | 0.264 | 0.136 | 0.240 | 0.311 | 0.034 | 0.045 |
| | CoT | 0.253 | 0.133 | 0.233 | 0.283 | 0.035 | 0.046 |
| | SR | 0.266 | 0.116 | 0.233 | 0.290 | 0.036 | 0.044 |
| | MAD | 0.241 | 0.127 | 0.249 | 0.261 | 0.038 | 0.046 |
| | DMAD | 0.243 | 0.141 | 0.239 | 0.285 | 0.036 | 0.046 |
| | CoThinker | 0.225 | 0.133 | 0.244 | 0.318 | 0.038 | 0.044 |

Table 11: Bootstrap Standard Error (SE) for all methods across model families, rescaled by 8B IO baseline. Values shown are rescaled SE (original SE divided by 8B IO baseline mean for each task). The small sampling variance (rescaled SE ranging from 0.034 to 0.318) demonstrates high reliability and stability of our bootstrap estimates.

## D  SYSTEM DESIGN AND IMPLEMENTATION DETAILS

### D.1  COTHINKER ARCHITECTURE DETAILS

#### D.1.1  COTHINKER AS PLUG-AND-PLAY CLT PRINCIPLES

CoThinker's primary contribution lies in providing **cognitive design principles** derived from Cognitive Load Theory (CLT) that can be integrated into existing multi-agent frameworks as "plug-and-play" enhancements. Rather than competing with established platforms, CoThinker offers a complementary cognitive lens that addresses a fundamental gap: most existing frameworks focus on engineering coordination without explicitly managing cognitive load.

The multi-agent LLM ecosystem encompasses diverse frameworks serving different purposes: **Engineering Platforms** like AutoGen (Wu et al., 2023) and CrewAI focus on conversational flows and role-based task delegation; **Domain-Specific Systems** like MetaGPT (Hong et al., 2023) and ChatDev (Qian et al., 2023b) target software development workflows; **Research Frameworks** like AgentVerse (Chen et al., 2023b) enable experimental exploration; **Workflow Systems** like LangGraph provide stateful execution; and **Consensus Systems** like Multi-Agent Debate (Du et al., 2023) and ReConcile (Chen et al., 2023a) focus on group reasoning.

Unlike these frameworks that primarily address engineering coordination, CoThinker introduces explicit cognitive load management through CLT-grounded design principles. Our approach provides three core cognitive enhancements: (1) dynamic thinking style assignment based on task demands rather than fixed functional roles, (2) communication moderation that balances cognitive similarity and diversity to prevent overload, and (3) transactive memory systems that enable cognitive offloading and specialization. These principles can augment existing frameworks—for instance, enhancing AutoGen's group chat with load-aware communication, replacing CrewAI's static roles with adaptive cognitive styles, or integrating TMS-based knowledge sharing into MetaGPT's development workflows.

### D.2  INFORMATION FLOW AND MESSAGE PASSING

This section provides detailed clarification of how information is generated and passed through the CoThinker system, addressing reviewer questions about system dynamics.

**Multi-Channel Information Architecture:** Each agent receives input through two distinct channels designed to manage cognitive load:

- **TMS Channel:** Provides high-level, synthesized, long-term context (group progress, consensus, "who knows what"), giving agents strategic awareness with low cognitive burden.
- **Communication Moderator Channel:** Provides specific, low-volume, current peer outputs for immediate reaction and refinement.

**Detailed Information Flow Example:** Consider Agent 1 in round $t$:

*Step 1: Information Gathering*

- Communication Moderator selects peers (e.g., Agent 2 & 3) based on cognitive distance and network topology
- Agent 1 accesses TMS to understand group consensus and peer specializations
- Agent 1 receives specific outputs from selected peers

*Step 2: Cognitive Processing*

- From TMS: "Agent 2 is performing fact-checking, Agent 3 is identifying mathematical formulas"
- Agent 1 realizes these tasks are covered, enabling cognitive offloading
- Agent 1 focuses on alternative solution approaches using Agent 3's formulas

*Step 3: Output Generation and System Update*

- Agent 1 generates new output informed by peer insights and TMS understanding
- All agent outputs update the TMS for next round
- Communication Moderator updates network topology based on new cognitive distances

**Network Dynamics:** The communication network reconstructs dynamically each round, with edges determined by:

- Fixed in-degree $N$ (CLT working memory constraint)
- Cognitive similarity for local clustering (reduces extraneous load)
- Probabilistic rewiring $\beta$ for diverse exploration (distributes intrinsic load)

This dual-channel, dynamic network approach distinguishes CoThinker from existing frameworks that typically use either broadcast communication or static network topologies.

### D.3 PROMPT ARCHITECTURE FOR AGENT PARALLEL THINKING

The Agent Parallel Thinking module in CoThinker aims to foster a beneficial division of cognitive labor by assigning diverse thinking styles to agents. This approach is grounded in theories of thinking styles, such as Sternberg's Theory of Mental Self-Government (Sternberg, 1997), which posits that styles are preferred ways of using one's abilities, not abilities themselves. This distinction is crucial: CoThinker leverages thinking styles as preferential orientations for LLM agents, assuming the base model possesses a broad set of underlying capabilities. The assigned style guides how these capabilities are applied to the task, rather than attempting to imbue a new, fixed skill or enforce a rigid behavioral script as a predefined "role" might. This aligns with findings that In-Context Learning often modulates an LLM's thinking style rather than altering its core knowledge (Lin et al., 2024; Zhao et al., 2025).

Adherence to a flexible thinking style is hypothesized to impose less extraneous cognitive load on an LLM agent compared to maintaining a complex, predefined role persona. This allows more of the agent's cognitive resources to be dedicated to the primary task. Furthermore, while core thinking styles are often seen as relatively stable, they are also understood to be somewhat malleable and can be adapted to specific task demands (Sternberg, 1997). CoThinker operationalizes this adaptability through a two-stage prompting strategy:

**1. Style Orchestration (**Orch **function):** The Thinking Style Orchestrator (itself an LLM) is provided with the overall task description $D$ and a Thinking Style Inventory. This inventory consists of base thinking styles derived from Sternberg's theory, encompassing dimensions such as Functions (Legislative, Executive, Judicial), Forms (e.g., Monarchic, Hierarchic), Levels (Global, Local), Scope (Internal, External), and Leanings (Liberal, Conservative). The Orchestrator's objective is to generate

a diverse yet task-relevant set of $M$ specific thinking styles $\{\phi_1, \ldots, \phi_M\}$, one for each agent $A_i$. For each agent, the Orchestrator takes one or a combination of Sternberg's dimensions as a base style $\psi_i$ and adapts it to the given task $D$. The Orchestrator is guided to ensure the resulting set of styles $\{\phi_i\}$ promotes varied perspectives on the problem, reflecting the value of different styles for different task facets.

An example prompt for the Orchestrator, given a base combination from Sternberg (e.g., $\psi_i$ = "Legislative-Global style"):

```
Given the primary task: "{Task D}"
And the base thinking style profile (from Sternberg's Theory of
Mental Self-Government): "{Base Style profile psi_i, e.g.,
Legislative function with a Global level preference}"

Generate a concise (1-2 sentences) task-specific adaptation
of this thinking style profile that would be most beneficial
for an agent contributing to this primary task. The agent
should focus its reasoning and output according to this
adapted style.
Task-Specific Style for an agent:
```

This process results in $M$ distinct, task-contextualized thinking styles $\{\phi_1, \ldots, \phi_M\}$. By dynamically adapting general styles to the specific task, CoThinker aims to harness the benefits of stylistic diversity while mitigating risks such as pigeonholing or oversimplification associated with static style assignments.

**2. Agent Instruction** (Agent **function - style incorporation**): Each agent $A_i$ then receives its specific thinking style $\phi_i$ as part of its instruction prompt, guiding its approach throughout the problem-solving process. An excerpt of an agent's prompt showing style incorporation:

```
You are Agent {num}. Your assigned thinking style for this
task is: "{Style phi_i generated by Orchestrator}".
The overall task is: "{Task D}".
[Other contextual information, e.g., from TMS mu^(t),
references P_i^(t-1), own previous thought x_i^(t-1)]

Keeping your assigned thinking style in mind, please provide
your thoughts/solution:
```

This method encourages agents to approach the problem from varied cognitive angles, promoting comprehensive exploration of the solution space and distributing the intrinsic cognitive load of the task, without the cognitive burden of strict role-playing.

### D.4    PROMPT ARCHITECTURE FOR TRANSACTIVE MEMORY SYSTEM (TMS) EMULATION

As introduced in Section 4.2, CoThinker incorporates a mechanism to emulate a human Transactive Memory System (TMS). A TMS is a collective cognitive resource developed by groups, encompassing a shared understanding of who knows what (metamemory or expertise directory), how to access and integrate this distributed knowledge, and a level of trust in the information provided by different members (Wegner, 1987; Hollingshead, 2001; Lewis, 2003). Effective TMS functioning involves processes of knowledge *encoding* (assigning information to members or recognizing expertise), *storage* (individuals retaining specialized knowledge), and *retrieval* (accessing and using the distributed knowledge), facilitated by member *specialization*, perceived *credibility*, and inter-agent *coordination* (Yoo & Kanawattanachai, 2001). This systematic division and integration of cognitive labor allows groups to handle more complex information and solve problems more effectively than individuals or less coordinated groups.

CoThinker's emulation of TMS centers on the generation and presentation of the collective memory state, $\mu^{(t)}$, at each round $t$. This is not merely an aggregation of past messages but a structured synthesis designed to reflect key TMS components. Specifically, an auxiliary LLM agent (the "TMS Manager") is tasked with populating a predefined "TMS Template" based on all agent outputs

$\{x_j^{(t-1)}\}_{j=1}^M$ from the previous round and the existing memory state $\mu^{(t-1)}$, to produce the updated $\mu^{(t)}$. This template explicitly guides the TMS Manager to synthesize information reflecting:

1. **Expertise Directory ("Who Knows What"):** The template prompts the TMS Manager to list the key contributions from each agent $A_j$ in the previous round, often implicitly linking these contributions back to their assigned thinking style $\phi_j$ or emergent problem-solving role. For example, $\mu^{(t)}$ might state: *"Agent A (Analytical Thinker) identified three inconsistencies in the data, while Agent B (Creative Ideator) proposed two novel solutions based on X."* This helps all agents maintain an updated awareness of which peer is focusing on, or has provided significant input regarding, specific facets of the task. This corresponds to the *encoding* of expertise and facilitates targeted *retrieval* cues.
2. **Shared Knowledge Store (Consensus and Artifacts):** The template requires the TMS Manager to identify and articulate points of emerging consensus, established facts, or partial solutions that the group has collectively built. For instance: *"Consensus: The primary bottleneck is resource allocation. Established: The budget cannot exceed Y."* This component of $\mu^{(t)}$ serves as the repository of *stored*, validated collective knowledge, reducing the need for agents to re-derive information and providing a foundation for subsequent reasoning.
3. **Differential Insights and Unresolved Issues (Focus for Coordination):** A crucial part of the TMS template prompts the TMS Manager to highlight discrepancies between agent outputs, unresolved questions, conflicting perspectives, or aspects of the problem that remain unaddressed. Example: *"Divergence: Agent C suggests strategy Alpha, while Agent D advocates for Beta. Unresolved: The feasibility of implementing X within the given timeframe."* This explicitly flags areas requiring further discussion, debate, or focused problem-solving in the next round, thereby guiding inter-agent *coordination* and ensuring that cognitive effort is directed towards the most critical, unresolved aspects of the task assigned to most relavent agents.

The structure of $\mu^{(t)}$, as generated by this templated process, is then presented to each agent $A_i$ at the beginning of round $t$ as part of its input prompt. An excerpt illustrating this presentation is:

```
[Agent's assigned thinking style: {Style_phi_i}]
[Overall Task: {Task_D}]

Collective Summary from Previous Round (reflecting shared understanding mu^(t)):
"{Text of mu^(t) generated by the TMS Manager using the TMS Template}"

Your Previous Output (x_i^(t-1)):
"{Text of x_i^(t-1)}"

Reference Outputs from Peers (P_i^(t-1)):
Reference 1 (from Agent A_k): "{Text of x_k^(t-1)}"
Reference 2 (from Agent A_l): "{Text of x_l^(t-1)}"
...

Based on all the above, and keeping your thinking style in mind,
provide your refined thoughts/contribution for the current round:
```

This deliberate structuring of $\mu^{(t)}$ to reflect an expertise directory, a shared knowledge store, and a pointer to unresolved issues distinguishes CoThinker's approach from simple multi-agent cooperation or discussion. While basic cooperation might involve information sharing, it often lacks the systematic assignment of knowledge domains, explicit tracking of expertise, and focused mechanisms for integrating specialized insights that a TMS provides. CoThinker's TMS emulation aims to create a more efficient and powerful "group mind" by embedding these principles directly into the information environment of the agents, thereby reducing redundant effort and enhancing the quality of collective problem-solving.

### D.5 COMMUNICATION MODERATOR: CULTIVATING AN EFFICIENT NETWORK VIA STRONG AND WEAK TIES

The transactional cost induced by inter-agent communication is a key concern in Collaborative CLT (Kirschner et al., 2009; 2018). The Communication Moderator in *CoThinker* (Section 4.3) strategically structures inter-agent communication by implicitly leveraging principles from social and complex network theories. This design fosters a network optimized for managing cognitive load and enhancing collective intelligence.

**Local Cohesion via Strong Cognitive Ties and High Clustering** The primary reference selection mechanism (with probability $1 - \beta$) connects agent $A_i$ to peers whose prior outputs $x_k^{(t-1)}$ are most cognitively similar to $A_i$'s own $x_i￼^{(t-1)}$. This promotes the formation of local clusters where agents process highly related information. From a social network perspective, these connections are analogous to **strong ties** (Granovetter, 1983), fostering cohesive subgroups. In network science, this behavior inherently leads to a high **local clustering coefficient**, indicating dense intra-group connectivity.

For similarity measurement, we employ cosine similarity of text embeddings (e.g., using `all-MiniLM-L6-v2`) to quantify cognitive proximity between agent outputs.

- **Rationale:** Such local clustering facilitates focused refinement of shared ideas and reduces the extraneous cognitive load associated with integrating highly similar information.

**Global Integration via Weak Cognitive Ties and Small-World Properties** Exclusive reliance on strong ties (i.e., $\beta = 0$) could lead to network fragmentation, where clusters become isolated "echo chambers." This corresponds to a lack of "bridging capital" across **structural holes** in social network theory (Burt, 2004), and a long **average path length** in network science, hindering the global distribution of diverse insights and the effective management of overall intrinsic cognitive load.

The probabilistic "rewiring" mechanism (with probability $\beta$) counteracts this by compelling agents to also reference randomly chosen peers, irrespective of immediate cognitive similarity.

- **Mechanism and Analogy:** These random connections function as **weak ties** (Granovetter, 1983), which are crucial for bridging disparate network segments and transmitting novel information.
- **Network Outcome:** Introducing such weak ties into a highly clustered network is a hallmark of **small-world networks** (Watts & Strogatz, 1998). These networks advantageously combine high local clustering with short global average path lengths.
- **Rationale:** In *CoThinker*, these $\beta$-driven connections ensure efficient propagation of diverse perspectives across cognitive clusters. This shortens the information path length, promotes the synthesis of varied knowledge, helps distribute the intrinsic cognitive load of the overall task, and prevents premature convergence.

In essence, the Communication Moderator dynamically cultivates a network with small-world characteristics. By balancing the formation of strong-tie local clusters for specialized processing with weak-tie bridges for global integration, it supports both deep, focused collaboration and the broad synthesis of diverse insights, crucial for effective collective problem-solving.

### D.6 SYNTHESIZER MODULE: CONSOLIDATION AND COGNITIVE GROUNDING

The Synthesizer module (Section 4.4) consolidates outputs from all agents ($\{x_i^{(T-1)}\}_{i=1}^M$) and the final Transactive Memory System state ($\mu^{(T-1)}$) into a single solution for the task $D$ after $T_{max}$ rounds. It can be implemented as an External Agent (dedicated LLM) or an In-group Agent (team member) (Lu et al., 2024; Shinn et al., 2023). The design choice for the Synthesizer can vary, with different cognitive implications drawing from Collaborative Cognitive Load Theory (Kirschner et al., 2018) and Observational Learning (Bandura & Walters, 1977):

1. **External Agent Synthesizer (Observational Learning):** This involves a dedicated LLM instance, distinct from the collaborating agents, to produce the final output. This agent receives all final individual perspectives and the collective memory summary.
   - *Cognitive Analogy:* This setup mirrors **Observational Learning** (Bandura & Walters, 1977). The External Synthesizer observes the diverse problem-solving behaviors and refined outputs

of the specialist agents. By analyzing these varied "models" of thought and their collective synthesis ($\mu^{(T-1)}$), it can construct a comprehensive solution, potentially integrating insights in a novel way without having been part of the iterative load distribution.

2. **In-group Agent Synthesizer (Collaborative Leading/Shared Regulation):** One of the existing collaborating agents (e.g., an agent identified as a leader or one with a consistently high-quality output, or a randomly chosen one) can be tasked with synthesizing the final solution. This agent uses its own understanding, the collective memory $\mu^{(T-1)}$, and the final outputs of its peers. align

   • *Cognitive Analogy:* This aligns with principles from **Collaborative Cognitive Load Theory (CCLT)** (Kirschner et al., 2018), specifically aspects of shared regulation and distributed leadership. The synthesizing agent, having participated in the collaborative process, leverages its deep contextual understanding and the established collective working memory ($\mu^{(T-1)}$) to guide the final integration. Its synthesis is an act of "collaborative leading" by taking responsibility for the final product based on the group's efforts.

**Sample Prompt for an External Agent Synthesizer** (Synth)**:**

```
Original Task: "[Task Description D]"
After collaborative thinking, the final individual
perspectives from M=[Number of Agents] agents are:
Agent 1: "[x_1^(T-1)]"
...
Agent M: "[x_M^(T-1)]"
The final collective understanding synthesized during
their collaboration is:
"[μ^(T-1)]"
Based on all this information, please generate a
comprehensive, high-quality, and coherent final
solution to the original task.
```

This prompt structure ensures the Synthesizer has all the necessary context to perform its role effectively.

# E  SUPPLEMENTARY EXPERIMENTAL DATA AND BENCHMARK DETAILS

## E.1  EXPERIMENTAL CONFIGURATION DETAILS

Here we give detailed configuration of experiments from Section 5.1.

**LLM API Parameters:** For all baseline methods (IO, CoT, SR) and the initial generation round ($t = 0$) of multi-agent methods (MAD, DMAD, *CoThinker*), the API temperature was set to "0.25" to encourage some diversity. For subsequent iterative rounds ($t > 0$) in *CoThinker*, MAD, and DMAD, the temperature was set to "0.0" and "frequency_penalty" to "0.5" to promote focused refinement and reduce repetition. Other API parameters (e.g., "top_p", "top_k") were left at their default values. Maximum output tokens were set to be large enough for each task, otherwise specified in the task description.

*CoThinker* **Default Configuration:** Unless specified otherwise in ablation studies, *CoThinker* used $M = 6$ agents, $T_{max} = 3$ interaction rounds (initial generation + 2 refinement rounds), a reference set size $N = 3$ (each agent receives messages from 3 peers), and an exploration rate $\beta = 0.3$.

All models run with the initial generation temperature set to 0.25 to encourage diverse outputs. In multi-agent settings, subsequent rounds use temperature 0.0 and a frequency penalty of 0.5 to reduce repetition. By default, multi-agent methods use $M$=6 agents interacting over $T$=3 rounds. For *CoThinker*, we set $N$=3 references and exploration parameter $\beta$=0.3.

## E.2  DETAILED BASELINE METHOD DESCRIPTIONS

The baseline methods used for comparison from Section 5.1 are implemented as follows:

- **Single Agent (Standard Prompt - IO):** The base LLM is given the task instruction directly, without any specialized prompting techniques, serving as a fundamental measure of its raw capability.
- **Single Agent (CoT):** Chain-of-Thought prompting (Wei et al., 2022) is employed, where the LLM is prompted to "think step by step" or provided with few-shot examples demonstrating a reasoning process before arriving at the final answer.
- **Single Agent (Self-Refine - SR) (Madaan et al., 2023):** This method involves an iterative process ($T = 3$ iterations). The LLM first generates an initial solution. Subsequently, it is prompted to critique its previous output and then to generate an improved version based on that critique.
- **Multi-Agent Debate (MAD) (Liang et al., 2023; Du et al., 2023):** Multiple LLM agents ($M = 6$) initially generate individual solutions. In subsequent iterative rounds ($T = 3$ total generations), each agent receives the solutions from all other agents from the previous round and is prompted to consider these peer solutions, critique them if necessary, and refine its own solution. The final answer is typically derived from the best-performing agent's output after the debate rounds.
- **Diverse Multi-Agent Debate (DMAD) (Liu et al., 2025b):** DMAD extends MAD by promoting diverse reasoning methods from the outset. Each agent is assigned a distinct prompting strategy (e.g., standard IO, Chain-of-Thought, Step-Back Prompting) to generate its initial solution, aiming to break "fixed mental sets." These diverse initial solutions are then shared and refined through iterative debate rounds, similar to MAD.

*Summary. (i) Single Agent (IO)* is a standard mode of prompting without additional techniques. *(ii) Single Agent (CoT)* incorporates Chain-of-Thought prompting to elicit step-by-step reasoning. *(iii) Single Agent (Self-Refine)* uses iterative self-critique and revision processes. *(iv) Multi-Agent Debate (MAD)* employs interactive agent discussion with consensus formation. *(v) Diverse MAD (DMAD)* introduces heterogeneous prompting to avoid fixed mental sets and encourage diverse perspectives.

### E.3 BENCHMARK DETAILS

**LiveBench (SOTA, comprehensive real-world tasks).** LiveBench (White et al., 2025) is a widely used 2025 SOTA benchmark featuring challenging, objectively scorable tasks (e.g., high school math competitions, zebra and word puzzles) that demand high cognitive load for humans. It integrates content from established benchmarks including Big-Bench Hard (Suzgun et al., 2023), AMPS (Hendrycks et al., 2021b), and IFEval (Zhou et al., 2023). Frequent updates minimize test data contamination so results reflect genuine reasoning rather than memorization. Covered domains include:

- *Mathematics:* competitive programming, olympiad-level math, algebraic simplification
- *Reasoning:* logical deduction and spatial reasoning
- *Language:* nuanced understanding and manipulation
- *Instruction Following:* adherence to complex instructions
- *Data Analysis:* structured data manipulation

The tasks are intentionally difficult—even strong models struggle—making LiveBench an ideal proxy for high cognitive load scenarios.

**CommonGen-Hard (controlled information interactivity).** CommonGen-Hard (Madaan et al., 2023), derived from CommonGen (Lin et al., 2020), systematically stresses element interactivity: the model must generate a coherent multi-sentence paragraph using 3–5 target concepts selected from a large pool (30) of distractors while maintaining narrative coherence and commonsense plausibility. We adopt a 10-dimensional evaluation metric (Li et al., 2018) and employ an LLM-based evaluator (Gemini-1.5-Pro), following the protocol in Ye et al. (2023) to mitigate LLM-as-judge bias. The evaluator uses a detailed rubric with per-dimension Likert (1–10) scoring and aggregates to a total score. The ten rubric dimensions are:

1. **Relevance to Query** (appropriateness and focus; highest weight)
2. **Conciseness** (brevity without loss of essentials)
3. **Clarity & Understandability** (ease of comprehension)
4. **Readability & Fluency** (natural flow; grammar)
5. **Comprehensiveness & Completeness** (covers all prompt aspects)

6. **Demonstrated Knowledge** (accurate commonsense/domain knowledge)
7. **Logic & Coherence** (internal consistency and structure)
8. **Originality & Creativity** (novel framing/ideas)
9. **Engagement & Interest** (compelling responses)
10. **Insightfulness & Depth** (analytical richness; lowest weight)

This setup provides a controlled environment to validate that our architecture manages the information interactivity predicted by CLT to cause performance degradation, while keeping evaluation fair and systematic.

### E.4 CROSS-MODEL DETAILED RESULTS

This section provides comprehensive cross-model evaluation results referenced in Section 5.4, including detailed performance breakdowns across multiple LLM families and task categories.

Table 12 presents the full cross-model evaluation results across all tested models and evaluation scenarios. Table 13 provides detailed task-specific results for the standard setting cross-model evaluation, showing CoThinker's performance across individual reasoning and mathematical tasks.

| Model | Method | Avg (SE) | Math (SE) | Reasoning (SE) |
|---|---|---|---|---|
| Gemini-2.5-Flash | CoThinker | **72.8 (2.67)** | **76.3 (2.57)** | **69.2 (4.15)** |
|  | DMAD | 59.7 (3.32) | 56.7 (4.48) | 62.8 (4.70) |
|  | IO (Baseline) | 45.1 (3.40) | 59.3 (3.78) | 31.0 (4.86) |
| GPT-4.1-Mini | CoThinker | **55.4 (2.92)** | **40.0 (3.72)** | **70.8 (3.90)** |
|  | DMAD | 39.1 (3.21) | 34.2 (3.84) | 44.0 (4.82) |
|  | IO (Baseline) | 37.4 (3.27) | 34.0 (3.91) | 40.8 (4.82) |
| Qwen3-32B | CoThinker | **22.1 (2.89)** | **18.9 (3.52)** | **25.2 (4.22)** |
|  | DMAD | 11.5 (2.38) | 8.8 (3.53) | 14.2 (3.25) |
|  | IO (Baseline) | 11.7 (2.33) | 3.4 (3.53) | 20.0 (3.14) |
| DeepSeek-R1-8B | CoThinker | **5.8 (2.28)** | 2.9 (1.42) | **8.8 (3.58)** |
|  | DMAD | 5.2 (1.71) | **3.8 (2.00)** | 6.5 (2.58) |
|  | IO (Baseline) | 2.3 (1.46) | 1.9 (1.94) | 2.8 (2.00) |

Table 12: Complete cross-model evaluation results on LiveBench under constrained setting (8192 tokens, greedy decoding). Values in parentheses are Bootstrap Standard Errors (SE) in percentage points. CoThinker demonstrates consistent and statistically significant improvements across diverse model families and capabilities.

| Model Configuration | Zebra Puzzle | Spatial | Math Comp | AMPS Hard |
|---|---|---|---|---|
| GPT-5-Nano (CoThinker) | 75.75 | 88.00 | 89.13 | 88.00 |
| GPT-5-Nano (IO) | 56.75 | 80.00 | 78.26 | 87.00 |
| Qwen3-30B-A3B (CoThinker) | 83.50 | 84.00 | 84.78 | 88.00 |
| Qwen3-30B-A3B (IO) | 79.00 | 76.00 | 73.91 | 77.00 |
| GPT-OSS-20B (CoThinker) | 55.00 | 80.00 | 78.26 | 82.00 |
| GPT-OSS-20B (IO) | 54.00 | 78.00 | 71.74 | 82.00 |

Table 13: Task-specific performance breakdown across models and tasks. Full CoThinker configuration consistently outperforms baseline single-agent approach across reasoning and mathematical tasks, demonstrating the effectiveness of the complete multi-agent architecture.

The detailed results reveal several key insights about CoThinker's generalization capabilities:

**Model Family Independence:** CoThinker demonstrates consistent improvements across architecturally diverse families (GPT, Gemini, Qwen, DeepSeek), indicating that the CLT-based approach generalizes beyond specific model architectures.

**Performance Scaling:** Stronger base models (e.g., GPT-5-Nano, Qwen3-30B-A3B) show larger absolute gains from CoThinker, suggesting that the framework effectively leverages higher baseline reasoning capabilities while providing substantial improvements for weaker models.

**Task-Dependent Benefits:** Mathematical reasoning tasks (math_comp, AMPS_Hard) and logical reasoning tasks (zebra_puzzle, spatial) consistently benefit from CoThinker's collaborative approach, validating the cognitive load distribution hypothesis for high-intrinsic-load tasks.

**Configuration Sensitivity:** The comparison between full CoThinker configuration and baseline single-agent approach shows that the complete multi-agent architecture (including TMS, Communication Moderator, and Style Generation) provides meaningful improvements over simplified configurations.

### E.5    Other Component Ablation Details

This section provides the complete component ablation results referenced in Section 5.6, showing detailed performance breakdowns for other model configurations tested. All configurations use M=6 agents and N=3 references with greedy decoding.

| Task | TMS: ON | TMS: OFF |
|---|---|---|
| Math: Math Comp | 35.6 | 7.7 |

Table 14: TMS ablation for Gemini-1.5-Flash-8B.

| Task | TMS: ON | TMS: OFF |
|---|---|---|
| Math: Olympiad | 47.1 | 44.2 |

Table 15: TMS ablation for Gemini-1.5-Flash.

| Task | TMS: ON | TMS: OFF |
|---|---|---|
| Data Analysis: CTA | 58.0 | 58.0 |
| Instruction Following: Paraphrase | 69.8 | 70.2 |
| Instruction Following: Simplify | 68.0 | 68.9 |
| Instruction Following: Summarize | 68.4 | 66.8 |
| Language: Connections | 53.5 | 52.6 |
| Reasoning: Zebra Puzzle | 48.2 | 39.5 |

Table 16: TMS ablation for Gemini-1.5-Pro.

**Analysis:** The TMS component shows variable effectiveness across model variants and task types. For the weaker Flash-8B model, TMS provides dramatic improvements on mathematical computation tasks, suggesting it helps manage cognitive load for complex reasoning. The standard Flash model demonstrates modest gains on advanced mathematical problems. The Pro model shows mixed results, with notable benefits for logical reasoning (Zebra Puzzle) but minimal impact on instruction-following tasks. This pattern indicates that TMS effectiveness depends on both model capability and task complexity, with greater benefits for cognitively demanding tasks on less capable models.

### E.6    Task-Wise Performance Data

*This subsection provides comprehensive raw scores for all subtasks across various model families and prompting methodologies, along with ablation studies investigating sensitivity to key hyperparameters.*

### E.7    Raw Subtask Performance Scores

The subsequent tables (Table 17 through Table 19) itemize the raw performance scores achieved on each subtask. Scores are reported to two decimal places. A hyphen (-) signifies missing or non-numeric data. Each table is dedicated to a distinct base model family.

| Subtask | IO | CoT | SR | MAD | DMAD | CoThinker |
|---|---|---|---|---|---|---|
| Connections | 13.50 | 18.17 | 17.33 | 17.67 | 17.00 | 19.33 |
| CTA | 54.00 | 50.00 | 30.00 | 48.00 | 52.00 | 54.00 |
| Math Comp. | 26.09 | 23.91 | 21.74 | 28.26 | 30.43 | 26.09 |
| Olympiad | 23.82 | 27.64 | 23.84 | 28.25 | 25.87 | 29.00 |
| Paraphrase | 74.27 | 72.82 | 38.42 | 65.22 | 66.55 | 46.02 |
| Simplify | 70.33 | 70.70 | 62.78 | 63.88 | 61.08 | 70.25 |
| Spatial | 34.00 | 28.00 | 18.00 | 34.00 | 22.00 | 28.00 |
| Story Gen. | 73.08 | 68.75 | 62.92 | 66.75 | 67.00 | 65.08 |
| Summarize | 69.35 | 71.27 | 50.43 | 58.32 | 62.62 | 42.32 |
| Table Join | 5.44 | 4.10 | 0.00 | 2.00 | 1.78 | 12.02 |
| Table Reformat | 80.00 | 82.00 | 36.00 | 38.00 | 50.00 | 60.00 |
| Zebra Puzzle | 16.00 | 22.25 | 17.25 | 22.75 | 17.00 | 25.75 |

Table 17: Raw scores for each subtask for Gemini-1.5-Flash-8B models across different prompting methods.

| Subtask | IO | CoT | SR | MAD | DMAD | CoThinker |
|---|---|---|---|---|---|---|
| Connections | 28.17 | 24.00 | 22.83 | 33.17 | 28.50 | 33.67 |
| CTA | 56.00 | 56.00 | 36.00 | 56.00 | 54.00 | 52.00 |
| Math Comp. | 41.30 | 39.13 | 39.13 | 41.30 | 41.30 | 41.30 |
| Olympiad | 32.20 | 34.37 | 33.35 | 34.41 | 33.27 | 36.89 |
| Paraphrase | 80.70 | 78.17 | 52.22 | 80.58 | 82.22 | 72.35 |
| Simplify | 75.83 | 77.68 | 67.57 | 72.07 | 74.40 | 69.00 |
| Spatial | 50.00 | 50.00 | 36.00 | 58.00 | 52.00 | 52.00 |
| Story Gen. | 76.25 | 77.50 | 57.92 | 60.75 | 80.75 | 79.50 |
| Summarize | 77.55 | 75.92 | 54.05 | 68.47 | 74.33 | 68.97 |
| Table Join | 21.64 | 22.78 | 8.12 | 15.00 | 32.60 | 31.20 |
| Table Reformat | 86.00 | 80.00 | 44.00 | 48.00 | 44.00 | 50.00 |
| Zebra Puzzle | 28.50 | 32.00 | 32.50 | 34.25 | 37.50 | 38.50 |

Table 18: Raw scores for each subtask for Gemini-1.5-Flash models across different prompting methods.

| Subtask | IO | CoT | SR | MAD | DMAD | CoThinker |
|---|---|---|---|---|---|---|
| Connections | 31.17 | 36.50 | 35.17 | 44.67 | 44.50 | 46.00 |
| CTA | 56.00 | 58.00 | 36.00 | 56.00 | 60.00 | 58.00 |
| Math Comp. | 47.83 | 36.96 | 45.65 | 54.35 | 56.52 | 56.52 |
| Olympiad | 51.79 | 54.77 | 50.16 | 59.63 | 58.46 | 62.72 |
| Paraphrase | 75.37 | 73.78 | 34.18 | 48.50 | 73.88 | 65.17 |
| Simplify | 74.77 | 75.72 | 54.48 | 55.43 | 72.88 | 66.37 |
| Spatial | 44.00 | 48.00 | 36.00 | 34.00 | 38.00 | 38.00 |
| Story Gen. | 69.72 | 68.05 | 42.55 | 56.85 | 67.30 | 73.05 |
| Summarize | 68.92 | 67.17 | 46.23 | 52.83 | 69.05 | 65.72 |
| Table Join | 35.98 | 32.56 | 16.16 | 43.82 | 42.32 | 44.18 |
| Table Reformat | 88.00 | 88.00 | 28.00 | 28.00 | 86.00 | 78.00 |
| Zebra Puzzle | 39.00 | 35.75 | 40.75 | 41.00 | 42.25 | 44.50 |

Table 19: Raw scores for each subtask for Gemini-1.5-Pro models across different prompting methods.

E.8   SUBTASK DESCRIPTIONS

The evaluation benchmark comprises a diverse array of subtasks, each designed to assess specific reasoning and generation capabilities of the models. Concise descriptions for each subtask category are provided below:

**Connections**: Assesses the model's aptitude for identifying and comprehending relationships (e.g., logical, causal, shared attributes) between disparate textual elements or conceptual ideas.
**CTA (Call to Action)**: Evaluates the model's effectiveness in generating or interpreting persuasive or directive language aimed at eliciting a targeted response or action.
**Math Comp. (Mathematical Computation)**: Measures the model's proficiency in executing mathematical calculations and resolving problems necessitating computational procedures.
**Olympiad**: Challenges the model with highly complex mathematical problems, characteristic of mathematics Olympiads, which demand profound reasoning and multi-step solution strategies.
**Paraphrase**: Tests the model's ability to accurately rephrase given text while preserving its original semantic content, thereby demonstrating linguistic understanding and versatility.
**Simplify**: Assesses the model's capacity to transform complex textual information into a more readily understandable format, typically by employing simpler vocabulary and sentence structures without loss of core meaning.
**Spatial**: Evaluates the model's spatial reasoning faculties, including its ability to understand and reason about objects in two or three-dimensional space, their interrelations, positions, and transformations.
**Story Generation**: Measures the model's creative ability to produce coherent, engaging, and contextually relevant narratives derived from specified prompts or constraints.
**Summarize**: Assesses the model's proficiency in condensing extended passages of text into succinct summaries that encapsulate the principal points and essential information.
**Table Join**: Evaluates the model's comprehension of relational data structures by requiring it to identify appropriate mechanisms for combining or linking multiple data tables based on common columns or keys.
**Table Reformat**: Tests the model's capability to manipulate tabular data by converting a table from one structural or data representation format to another, adhering to provided instructions.
**Zebra Puzzle**: Assesses the model's deductive reasoning and constraint satisfaction abilities through logic puzzles (such as Einstein's Puzzle) that necessitate deriving a solution from a given set of clues.

E.9   ABLATION STUDY: IMPACT OF REFERENCE SET SIZE (N)

This study investigates the influence of varying the reference set size (hyperparameter N) on model performance across selected subtasks. N dictates the number of prior examples or "thoughts" considered by the model during generation. Values of N from 0 (representing a baseline, e.g., standard CoT where N/A) to 5 were evaluated using the Gemini-1.5-Flash-8B model. The results are illustrated in Figure 5.

**Analysis of Figure 5:**

- The general trend in performance on these reasoning-intensive ('olympiad', 'spatial', 'zebra_puzzle') and language-based ('connections') tasks is examined to determine if it improves, plateaus, or reveals an optimal N value.

- Performance at N=0 (baseline) is contrasted with N>0 configurations to ascertain whether the introduction of a reference set confers a tangible advantage for these specific tasks.

- The differential sensitivity of subtasks to variations in N is analyzed, particularly for computationally demanding tasks like 'olympiad' (Math) or 'zebra_puzzle' (Reasoning) relative to 'connections' or 'spatial'.

- The investigation seeks to identify if a particular N value (e.g., N=2 or N=3) consistently yields superior scores or an advantageous performance-cost balance across these subtasks.

- Evidence for diminishing returns is sought, where increasing N beyond a certain point might lead to marginal gains or even performance degradation, potentially due to the introduction of noise or distracting elements from an overly large reference set.

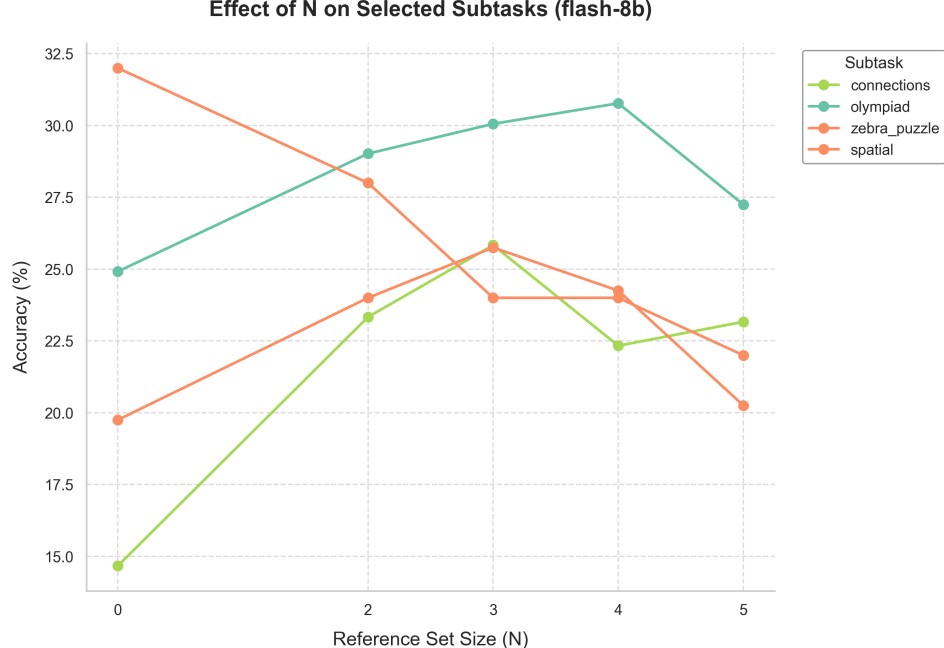

Figure 5: Effect of Reference Set Size (N) on performance for selected subtasks ('connections', 'olympiad', 'zebra_puzzle', 'spatial') using the Gemini-1.5-Flash-8B model. Subtasks are color-coded by their primary category.

*Contextual Note:* Reasoning and mathematical tasks are often hypothesized to benefit from a moderately sized, diverse reference set. While N=0 or N=1 might provide insufficient context, excessively large N values could introduce irrelevant information.

### E.10   ABLATION STUDY: IMPACT OF EXPLORATION RATE (BETA)

This ablation study explores the effect of the exploration rate (hyperparameter Beta) on model performance for selected subtasks, maintaining a fixed reference set size of N=2. Beta influences the diversity of thoughts or solutions generated by the model. The Gemini-1.5-Flash-8B model was employed for this analysis (Figure 6).

**Analysis of Figure 6:**

- The analysis aims to identify an optimal or effective range for Beta where performance peaks for the selected subtasks, which include data analysis ('tablejoin'), instruction following ('story_generation', 'simplify'), and mathematical computation ('math_comp').

- The impact of extreme Beta values (both very low, indicating minimal exploration, and very high, indicating extensive exploration) on performance is examined for potential suboptimality.

- Differential responses to Beta across subtasks are investigated, for instance, whether creative tasks like 'story_generation' benefit from a different Beta regime compared to more structured tasks such as 'math_comp' or 'tablejoin'.

- The stability of performance across the spectrum of Beta values is assessed, noting any significant fluctuations versus relatively consistent scores within particular ranges.

*Contextual Note:* A moderate Beta value (e.g., 0.3-0.6 in analogous systems) often represents a balance. Excessively low Beta values might risk premature convergence on suboptimal solutions, while overly high values could lead to an excessively diverse, and potentially lower-quality, set of outputs.

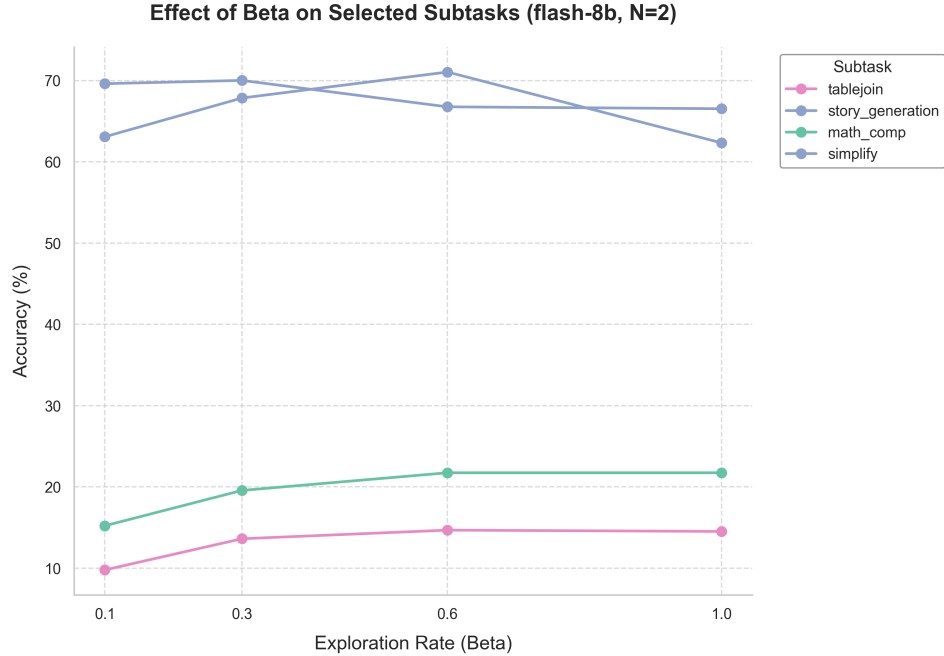

Figure 6: Effect of Exploration Rate (Beta) on performance for selected subtasks ('tablejoin', 'story_generation', 'math_comp', 'simplify') using Gemini-1.5-Flash-8B with N=2. Subtasks are color-coded by their primary category.

### E.11 ABLATION STUDY: IMPACT OF NUMBER OF AGENTS (M)

This study assesses the influence of the number of agents (hyperparameter M) on performance across all subtasks, with the reference set size fixed at N=3. M denotes the number of independent reasoning paths or "thinkers" utilized by the model. The Gemini-1.5-Flash-8B model was used for this evaluation (Figure 7).

Figure 7: Effect of Number of Agents (M) on performance across all subtasks for Gemini-1.5-Flash-8B with N=3. Each facet corresponds to a subtask, color-coded by its primary category.

**Analysis of Figure 7:**

- The overall impact of increasing M on performance is analyzed to determine if it generally leads to improvements across most subtasks or if the effects are heterogeneous.
- A cost-benefit perspective is considered, as higher M values, while potentially enhancing performance, also incur increased computational overhead. The study seeks an M value that offers a good trade-off.
- Subtasks that derive particular benefit from a larger number of agents are identified; for example, complex reasoning tasks or those requiring diverse perspectives might exhibit more substantial gains.
- The analysis looks for a saturation point where the benefits of increasing M diminish or where performance might even degrade for some (or all) tasks.

*Contextual Note:* Employing a greater number of agents can enhance the robustness and breadth of exploration. However, an excessive number might not yield significant incremental value or could potentially introduce noise if the aggregation of outputs from multiple agents is not optimally managed.

### E.12    ABLATION STUDY: PERFORMANCE FOR SPECIFIC M/N

This analysis evaluates performance across three distinct (M, N) configurations for the Gemini-1.5-Flash-8B model: M6_N3, M12_N6, and M18_N3. These evaluations are conducted under the "With Style" configuration, with Beta fixed at 0.3 and T (temperature or trials) at 3. Results are presented in Figure 8.

**Subtask Perf. M/N Configs (With Style, B=0.3, T=3, gemini-1.5-flash-8b): Performance on All Subtasks**

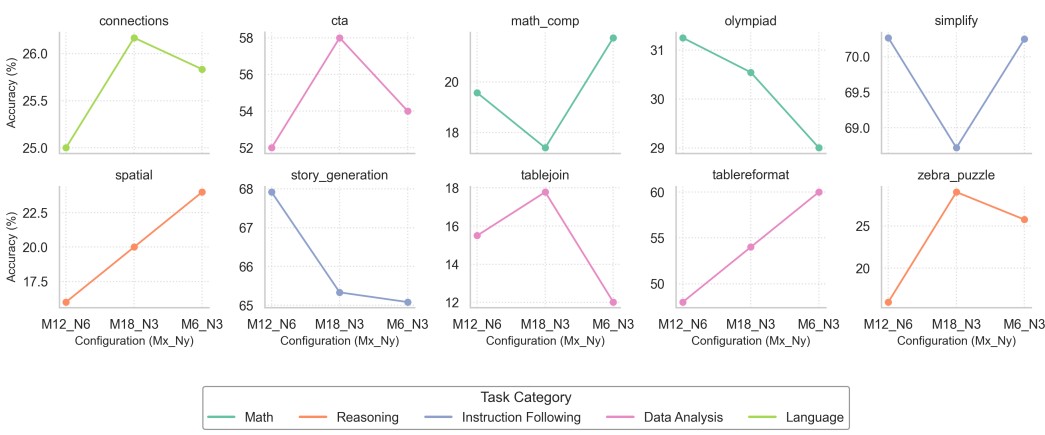

Figure 8: Subtask performance for specific M/N configurations (M6_N3, M12_N6, M18_N3) using Gemini-1.5-Flash-8B under the configuration (Beta=0.3, T=3). Faceted by subtask.

**Analysis of Figure 8:**

- The investigation aims to identify which of the tested (M, N) pairs yields the most favorable performance, either broadly across subtasks or for specific, critical subtasks.
- The trade-off between computational cost and performance gain is considered, as the configurations (M6_N3, M12_N6, M18_N3) entail different computational demands.
- The interaction between M and N is observed by comparing configurations; for instance, whether simultaneous increases in M and N (e.g., M6_N3 to M12_N6) lead to consistent improvements. The M18_N3 configuration provides insight into a different scaling strategy (higher M, moderate N).
- Consistency in the ranking of these (M, N) configurations across different subtasks is examined.

*Contextual Note:* This study assists in identifying potentially effective fixed configurations by exploring varied scaling strategies for the hyperparameters M and N within the "With Style" framework.

# F    LIMITATIONS AND FUTURE WORK

Our work successfully demonstrates the utility of Cognitive Load Theory (CLT) as a generative framework for designing multi-agent LLM systems. The limitations of this initial study are best understood as defining the current boundaries of our methods and charting a course for developing more robust tools for this new area of research.

**Toward Runtime Cognitive Load Measurement and Adaptive Systems.**    A central challenge is developing better **runtime measurement** capabilities for cognitive states in LLMs. Our current proxies (attention entropy and perplexity) require ground-truth answers, making them valuable for design-time validation but unsuitable for runtime adaptation. To enable truly adaptive systems that detect cognitive overload during inference, three key quantities require measurement:

*(1) Working Memory with Chunking:* Understanding how LLMs organize and chunk information into representational units—what constitutes an "element" in their working memory beyond simple token counts.

*(2) Dynamic Cognitive Load During Inference:* Tracking how cognitive load transforms through reasoning chains, as recent work shows attention patterns evolve with reasoning steps (Wen et al., 2025b; Li et al., 2023c). Attention-based runtime monitoring offers potential solutions.

*(3) Metacognitive Sensitivity:* Calibrating uncertainty signals to distinguish processing difficulty from knowledge gaps. Recent advances in attention-based uncertainty quantification (Li et al., 2025) and RL-based uncertainty training (Damani et al., 2025) provide promising directions.

Once these components are quantified, our validated proxy framework provides the foundation for runtime prediction systems that can adaptively decide when to invoke multi-agent collaboration—a critical step toward truly intelligent cognitive load management.

**Developing Universal Proxies for Cognitive Load.**    Beyond runtime measurement, we need more **universal proxies** for design-time analysis. Our current measures can be influenced by model architecture and tokenization. A critical direction is developing standardized "cognitive toolkits" that work reliably across diverse model families, potentially including gradient-based sensitivity analysis or direct elicitation methods as suggested by reviewers.

**Boundary Conditions and Task-Adaptive Collaboration.**    Our findings help delineate the **boundary conditions** under which CLT-based collaboration helps. Benefits are most pronounced for high intrinsic cognitive load tasks. Future work should develop principled methods to predict *a priori* which tasks require collaborative architectures, potentially through cognitive load estimation before execution. This connects to the adaptive system vision above—determining not just *when* to collaborate during a task, but *which tasks* benefit from collaboration at all.

**Emergent Collective Dynamics.**    While *CoThinker* operationalizes mechanisms for collective cognition, the rich internal social dynamics remain underexplored. Applying methods from **computational social science** to analyze interaction patterns—network evolution, transient leadership, consensus formation—could reveal deeper insights into artificial collective intelligence and potentially inform improved architectures.

**Human-AI Collaboration.**    A particularly exciting direction is extending the framework to include human users as specialized agents, creating **human-AI cognitive systems** where the architecture actively manages cognitive load for both human and AI participants. This could enable solving problems that neither could tackle alone, fostering true hybrid intelligence.

**Bidirectional Benefits: LLMs as Cognitive Science Research Tools.**    Finally, we note the potential for **bidirectional knowledge transfer**. While we apply cognitive science to improve LLMs, studying LLMs offers more controllable experimental paradigms than human studies. Investigating how LLMs chunk information, manage cognitive load, and develop collective intelligence could yield new insights about cognitive mechanisms that inform human cognitive science itself.

Addressing these future directions will advance our understanding of how to build truly collaborative and cognitively capable LLM-based systems while potentially contributing back to cognitive science through computational modeling.

