# OpenReview forum: "United Minds or Isolated Agents? Exploring Coordination of LLMs under Cognitive Load Theory"
_ICLR.cc/2026/Conference — ICLR 2026 Conference Desk Rejected Submission_

### Official Review · Reviewer_Vnt8 · 2025-10-22

**Soundness:** 3
**Presentation:** 3
**Contribution:** 1
**Rating:** 2
**Confidence:** 4

**Summary:**

This paper proposes a new multi-agent prompting framework for enhancing LLM reasoning particularly in complex reasoning problems; The paper draws analogy between In-context learning failures in LLMs and cognitive load theory, claiming that the ICL's limitaiton can be viewed as a cognitive overload where the working memory of the LLM is not enough for the task's cognitive load. The paper then proposed CoThinker, which divides the cognitive loads to multiple parallel agents and maintains a shared working memory. The CoThinker framework shows moderate improvements on LiveBench and CommonGen-Hard compared with direct IO, CoT and previous multi-agent debating methods.

**Strengths:**

- The analogy from cognitive load theory is interesting, and has the potential to open up new perspectives in understanding and improving large language models
- A comprehensive pilot study and citations from the cognitive science perspective is provided
- The empirical results show that the framework generalize well on different llms
- The paper is clearly written and easy to follow

**Weaknesses:**

- Although the paper spend a large portion trying to justify the CLT-LLM analogy, my main concern are as follows:
  - It is still not clear what does the "cognitive load" mean in an LLM: what's the difference between a "cognitive load" and simply "task difficulty"/"reasoning complexity"; both the attention entropy and perplexity only reflects the "uncertainty" of the model, which is also directly correlates with the difficulty/complexity of the task.
  - It is not clear why the analogy helps: I find it hard to justify any unique insights (compared with existing work on prompting methods) that the CLT-LLM analogy can bring. The final proposed method still boils down to: a central meta-agent + multiple sub-agents. I feel the analogy does not truly contribute to proposing a novel and effective method.
  - And on LiveBench, the gains against the second best baseline is very marginal; again posing a challenge on why this analogy helps
- Some very related work are missing in both related work and experiments:
  - Wang, Zhenhailong, et al. "Unleashing the emergent cognitive synergy in large language models: A task-solving agent through multi-persona self-collaboration." arXiv preprint arXiv:2307.05300 (2023).
  - Suzgun, Mirac, and Adam Tauman Kalai. "Meta-prompting: Enhancing language models with task-agnostic scaffolding." arXiv preprint arXiv:2401.12954 (2024).

**Questions:**

N/A

---

> ### Author Response · Authors · 2025-11-19
>
> Thank you for your thoughtful and constructive review. We address the main concerns in the following order: (1) We clarify Cognitive Load analogy and how it differs from "task difficulty"; (2) We demonstrate the non-obvious, predictive patterns in our empirical results that validate CLT; (3) We discuss the relationship to suggested related works.
>
> **Our core idea**: CLT is not merely a descriptive analogy but a principle-guided framework that provides falsifiable predictions and novel design guidance for multi-agent systems, distinguishing our approach from heuristic engineering.
>
> ---
>
> ## **1. Response to W1: The Value and Novelty of the CLT-LLM Analogy**
>
> We organize this response into three subsections addressing the reviewer's concerns about conceptual clarity (1.1), measurement validity (1.2), and architectural novelty (1.3).
>
> ### **1.1. Cognitive Load is Distinct from Task Difficulty**
>
> We clarify "task difficulty" or "reasoning complexity" are different from "cognitive load".
>
> "Task difficulty" and "reasoning complexity" are external properties of a problem which reflected by the knowledge and reasoning steps need to know to solve the problem.  Cognitive load is an internal processing state that depends on both the task and the agent itself. Critically, the same task can induce different cognitive loads depending on how one agent attempts to solve it.
>
> Here we present an example where high difficulty problem can leads to both high and low cognitive load for an agent. Consider a complex mathematical problem which an weak agent is not capable to solve.
>
> (1) In first scenario, the agent may continue reasoning and reflection, go back and forth, stuck in a loop and cannot give an answer. This creates **high cognitive load** for this agent making it unable to give a certain answer.
>
> (2) In second scenario, if you force the agent to give an answer, the agent may hallucinate a seemingly correct answer with confidence resulting a **low cognitive load** situation.
>
> Both scenario have the same difficult task which agent cannot answer correctly but result in deferent cognitive load. This distinction matters because it shifts our focus from observing model failures on hard tasks to understanding why they fail.

---

> ### Author Response · Authors · 2025-11-19
>
> ### **1.2. Our Proxies Measure Cognitive Mechanisms Beyond Simple Uncertainty**
>
> We clarify that these measures probe specific cognitive mechanisms under our controlled experiments, grounded in established cognitive science constructs.
>
> **Attention Entropy as a Proxy for Cognitive Load:** First, we clarify that, though we use the term "attention entropy", it is a metric showing sparsity rather than uncertainty. To understand what attention entropy measures, we must first clarify the concept of working memory. In cognitive science, working memory refers to the capacity to simultaneously hold and manipulate a limited number of **elements** [1,2]. A high cognitive load task requires using more element simultaneously to solve.
>
> In LLMs, attention entropy directly measures how much "element" is used simultaneously for prediction. When attention is scattered uniformly across many tokens(high entropy), it suggests the model have to use all in context information simultaneously to predict the next token, corresponding to high cognitive load. Additionally, this interpretation is supported by recent theoretical work showing that  Chain-of-Thought create sparse attention patterns by introducing sequential dependencies that naturally chunk information and thus enhancing performance [3,4,5].
>
> Our experiments also validate this mechanism beyond simple correlation with task difficulty; We find non-obvious result that reasoning leads to sparser attention even when context is longer. Mathematically, for N attention weights whose $E[a_i]=1/N$, attention entropy should increase as $H \sim \log N$. Thus naturally, longer sequence should have higher attention entropy. However, Table 6 shows that adding reasoning steps, actually decreases attention entropy (e.g. Level 3: 5.043->4.937). This non-trivial finding demonstrates that attention entropy reflects cognitive load.
>
> **Perplexity as a Conditional Proxy:** We clarify that perplexity relates to uncertainty but it doesn't simply correlate with the task difficulty. Additionally, using perplexity as proxy is supported by previous cognitive science research.
>
> In metacognitive theory, processing fluency (confidence), the cognitive ease, correlates with task difficulty only when the person is not cognitive overloaded; Under cognitive overload, one lose its meta-cognition to evaluate its confidence and may lead to confident errors (where confidence don't predict difficulty) [6,7].
>
> Two evidence shows that perplexity as proxy does reveal above phenomenon (with and without cognitive overload). (1) Generally speaking, in our controlled experimental (Section 3.2) setting where a question/answer pair is provided (agent do not reason by itself). We do find harder tasks corresponds to higher perplexity. (2) When the agent reason by itself, perplexity has almost no correlation with task performance especially in long context scenario [8] (as cognitive overload happens). Therefore these evidence captures CLT mechanism that validates perplexity as a both theoretically and empirically sound proxy
>
>
> ### **1.3. How CLT Generates Novel and Principled System Design**
>
> We clarify that the innovation lies not in inventing new modules but in how CLT principles guide their design, moving from ad-hoc heuristics to falsifiable, theory-driven choices. Our components—Transactive Memory System (TMS) and Communication Moderator (CM)—are established constructs from research on human collective cognition [9,10] in managing cognitive load, describing emergent behaviors in social and cognitive psychology. Rather than engineering arbitrary solutions, we operationalize these CLT-derived principles into computational modules.
>
> This principle-guided approach generates falsifiable predictions that distinguish our framework from heuristic baselines. For example, CLT predicts that collaboration benefits should be task-dependent as shown in our ablation study. Furthermore, CLT also explain how certain components help in our post-experiment analysis, say, (1) PPL decrease when thinking style and TMS are enabled (align with what CLT predicts); (2) Diverse peer answers increase PPL (same as what CLT predicts, diversity is good in exploring different answers but can add potential extraneous cognitive load). Detailed results can be found in table 13, 14, 15.
>
> ---

---

> ### Author Response · Authors · 2025-11-19
>
> ## **2. Response to W1: Empirical Validation Through Predictive Patterns**
>
> While Table 2 (main text) shows modest gains on Gemini-1.5 models, Table 8 (Appendix E.3) reveals substantial improvements on newer models. For Gemini-2.5-Flash, CoThinker achieves 76.3 vs. 56.7 on Math (+34.6% over DMAD) and 69.2 vs. 62.8 on Reasoning (+10.2%). For GPT-4.1-Mini: 40.0 vs. 34.2 on Math (+17.0%) and 70.8 vs. 44.0 on Reasoning (+60.9%). Qwen3-32B (22.1 vs. 11.5 average) show consistent improvements.
>
> This pattern suggests that newer models benefit more from our framework, which we interpret as evidence that recent models exhibit stronger human-like social cognition capabilities. Since our framework operationalizes CLT principles through LLMs' emergent social behaviors—their ability to adopt thinking styles, maintain collective memory to understand others, and engage in structured communication—models with better-developed social-cognitive abilities naturally enable our architecture to function more effectively. This observation opens an intriguing avenue: as models continue to develop more human-like emergent behaviors, CLT-grounded architectures may become increasingly powerful.
>
> ---
>
> ## **3. Response to W2: Relationship to Suggested Related Works**
>
> We thank the reviewer for highlighting these essential works. We have incorporated them into our revised **Related Work** and **Appendix** to ensure a comprehensive literature review.
>
> Analysis of Suggested Works:
> Both **Wang et al. (2023)** (*Multi-persona Self-collaboration*) and **Suzgun & Kalai (2024)** (*Meta-prompting*) represent a significant advancement in enhancing single-model reasoning. They share a common core strategy: decomposing complex tasks into sub-tasks handled by diverse "personas" or "experts" within a single LLM, orchestrated for hierarchical reasoning.
>
> **Connection to Our Baselines:**
> Crucially, our experimental baseline, **Diverse Multi-Agent Debate (DMAD) (Liu et al., 2025b)**, is a state-of-the-art extension built upon the concepts of multi-persona collaboration above above two paper. DMAD breaks "fixed mental sets" by enforcing diverse initial reasoning paths—conceptually aligning with Wang et al.'s approach. Therefore, our experiments comparing CoThinker against DMAD (Table 2 & 8) effectively benchmark our performance against the advanced strategies represented by these suggested works.
>
> **Our Unique Contribution:**
> While these works successfully demonstrate *that* diversity and decomposition improve performance (engineering view), our work provides the **cognitive mechanism** explaining *why* they work (scientific view)—specifically, by distributing *intrinsic cognitive load* to prevent working memory overload. Beyond explanation, we introduce **principled architectural constraints** (e.g., limiting peer connections via $N=3$ to respect WM limits, unlike fully connected debates) that optimize these systems, moving from heuristic prompt engineering to theory-guided design.
>
> ---
>
> **References:**
>
> [1] Baddeley, A. "Working memory." *Science* 255(5044):556-559 (1992).
>
> [2] Cowan, N. "The magical number 4 in short-term memory: A reconsideration of mental storage capacity." *Behavioral and Brain Sciences* 24(1):87-114 (2001).
>
> [3] Li, Yingcong, et al. "Dissecting chain-of-thought: Compositionality through in-context filtering and learning." *Advances in Neural Information Processing Systems* 36 (2023): 22021-22046.
>
> [4] Wen, Kaiyue, et al. "From Sparse Dependence to Sparse Attention: Unveiling How Chain-of-Thought Enhances Transformer Sample Efficiency." *The Thirteenth International Conference on Learning Representations* (2025).
>
> [5] Cui, Yingqian, et al. "A Theoretical Understanding of Chain-of-Thought: Coherent Reasoning and Error-Aware Demonstration." *Proceedings of The 28th International Conference on Artificial Intelligence and Statistics*, PMLR 258:3475-3483 (2025).
>
> [6] Alter, Adam L., and Daniel M. Oppenheimer. "Uniting the tribes of fluency to form a metacognitive nation." Personality and social psychology review 13.3 (2009): 219-235.
>
> [7] Koriat, A. "The feeling of knowing: Some sources of feelings of knowing and not knowing." In *The Oxford Handbook of Metamemory*, pp. 213-287. Oxford University Press (2012).
>
> [8] An, C., et al. "What is Wrong with Perplexity for Long-context Language Modeling?" *arXiv preprint* (2024).
>
> [9] Zhang, Jintian, et al. "Exploring Collaboration Mechanisms for LLM Agents: A Social Psychology View." *Proceedings of ACL* (2024).
>
> [10] Kirschner, Paul A., et al. "From cognitive load theory to collaborative cognitive load theory." *International Journal of Computer-Supported Collaborative Learning* 13:213-233 (2018).

---

> ### Comment · Reviewer_Vnt8 · 2025-11-24
>
> I thank the authors for providing detailed explanations; However, I have a few concerns regarding the author response:
>
> - 1.1: The author explains that the cognitive load is "an internal processing state that depends on both the task and the agent itself." This is not consistent with the submission where the cognitive load is defined as a **"task's cognitive load" (Line 56);** and is mentioned multiple times throughout the paper. And the agent-side's capacity is defined as the Working Memory (WM). I feel the authors have inconsistent definition comparing the submission and the rebuttal.
>
> - 1.3  Existing multi-agent frameworks are also **inspired** from multiple cognitive theories (cognitive synergy; role-playing; theory-of-mind; etc) and are definitely **not "engineering arbitrary solutions"**. I think we **should be very cautious when we claim that a cognitive theory "supports" a machine learning idea (as its theoretical proof).** This is a claim that is very hard to find sufficient evidence given our weak understanding of both the human brain and the large neural models (needless to say their drastic difference).
> The reason I ask the question is that I don't think the empirical results is enough for *supporting* this big claim, and I also don't see a significant contribution empirically on the final implementation of the method *inspired* by the cognitive view.

---

> > ### Author Response · Authors · 2025-11-24
> >
> > We thank the reviewer for the careful reading and acknowledge both points deserve clarification.
> >
> > **Re: 1.1 (Definitional Consistency)**
> >
> > This reflects an imprecision in our writing rather than a conceptual confusion. To clarify our intended framework:
> >
> > - Cognitive Load (CL) is indeed an interaction between task demands and agent capacity—not a fixed property of either alone. The phrase "task's cognitive load" at Line 56 is shorthand for "the cognitive load imposed by the task on a particular agent." due to page limits.
> >
> > - This aligns with the original CLT formulation (Sweller, 2011): intrinsic load depends on element interactivity relative to learner expertise. The same task imposes different loads on different agents.
> >
> > We will revise the related passages to read: "the cognitive load that a task imposes on an agent" to eliminate ambiguity.

---

> > > ### Author Response · Authors · 2025-11-25
> > >
> > > **Re 1.3**
> > >
> > >
> > >
> > > We want to  clarify that, your mentioned related works, for example, cognitive synergy is a valuable and insightful framework—it provides principled understanding of how diverse agents collaborate effectively, and its core ideas are also utilized in our baseline DMAD. In fact, cognitive synergy insights can inspire some aspects of our design (diverse thinking),

---

> > > > ### Author Response · Authors · 2025-11-25
> > > >
> > > > **Regarding "CLT as support":**
> > > >
> > > > We are not sure what do you mean by questioning *"cognitive theory  "supports" a machine learning idea"*—if cognitive theories cannot "support" ML methods, then how do cognitive synergy, Theory of Mind, and role-playing theories support their respective multi-agent frameworks? You acknowledged these are "principled" approaches, which implies these theories do provide valid support at the behavioral level.
> > > >
> > > > Our work follows the same logic, but goes one step further: beyond behavioral level support, we demonstrate functional-level support through our proxies (attention entropy, perplexity). Table 6 shows LLM attention patterns align with CLT predictions about information processing—this is evidence at the functional level, not just behavioral outcomes.
> > > >
> > > > We appreciate the your reminder to be precise in our language. As for the "big claim", we will be more cautious to define our works scope -- behavioral level and functional level, but not claiming internal equivalence of human and LLM agent.

---

> ### Author Response · Authors · 2025-11-25
>
> We sincerely appreciate you engaging in this discussion and reading our paper so thoroughly, and we hope these detailed responses resolve your concerns. We firmly believe that "**Well-defined social science theory can inspire machine learning research**" and that translating these principles into effective systems is a meaningful goal. We wish to move the art of LLM collaboration closer to a science with explicit guidance from well-defined theory which can help developers design their system more easily. As Donald Knuth noted, "**Science is what we understand well enough to explain to a computer.**"

---

> > ### Author Response · Authors · 2025-11-26
> >
> > Dear Reviewer Vnt8, We have updated the PDF to clarify the cognitive load definition (1.1), add discussion on related cognitive theories (1.3), and refine our theoretical claims. Could you please confirm if our reply and revision resolve your concerns? Thank you for the discussion.

---

> > > ### Comment · Reviewer_Vnt8 · 2025-11-27
> > >
> > > I thank the authors for providing additional clarifications and for revising the paper to prevent confusions on the definition of cognitive load. I will raise score to reflect these efforts. However I still have concerns on overclaiming; I would encourage the authors to further make sure the claims are well scoped.

---

> > > > ### Author Response · Authors · 2025-11-28
> > > >
> > > > Dear Reviewer Vnt8,
> > > >
> > > > Thank you for confirming that our clarifications and revisions regarding the definition of cognitive load and related theories have addressed your primary concerns. We will follow your advice and further ensure our theoretical claims are appropriately scoped in the final manuscript to avoid any perception of overclaiming.

---

### Official Review · Reviewer_XuKf · 2025-10-22

**Soundness:** 2
**Presentation:** 2
**Contribution:** 2
**Rating:** 4
**Confidence:** 4

**Summary:**

This paper attempts to connect cognitive load theory with limitations of large language models, and builds a multi-agent framework to mitigate cognitive overload through agent specialization and structured communication.

**Strengths:**

1. The identified issue about the mismatch between task complexity and model processing capability is important and deserves further exploration.
2. The experiments are comprehensive, with solid ablation studies and well-organized analyses.

**Weaknesses:**

1. Cognitive science is mainly used as a rhetorical framing to justify a fairly standard multi-agent architecture (e.g., role assignment, communication bus, small-world topology). While it works, the CoThinker system lacks real novelty in design. I would expect cognitive theories to inspire genuinely new forms of multi-agent organization, rather than merely serving as interdisciplinary justification for existing designs.
2. If CLT is to be meaningfully applied to LLMs, the key questions should be: how to measure a model’s working-memory capacity, how to quantify a task’s cognitive load, and most crucially, how to determine (in a quantifiable way) tasks be decomposed or allocated once both cognitive load and working-memory capacity are measurable. The paper does not directly address these points.
3. The validation experiments in Section 3 do not provide real evidence; they merely restate an obvious fact: harder tasks make the model less confident, and clearer instructions help with difficult problems.

**Questions:**

1. How do you envision quantitatively measuring “working memory capacity” in LLMs, beyond indirect proxies like attention entropy or perplexity?
2. Could the proposed framework adaptively estimate cognitive load and decide when to invoke multi-agent collaboration?
3. How sensitive is the performance of CoThinker to the chosen communication topology?

---

> ### Author Response · Authors · 2025-11-17
>
> We thank the reviewer for the thoughtful and insightful comments. In this rebuttal, we first address **W2 and related questions (Q1, Q2)** regarding direct quantification and our framework's approach, then clarify **W1** on our theoretical contributions, followed by **W3** on the pilot study findings, and finally **Q3** on communication topology sensitivity.
>
> ---
>
> ## **Clarification to W2 (and related Q1, Q2): "Not directly addressing: how to measure WM, quantify CL, determine task decomposition"**
>
>  **Thank you for raising this important question. We would like to clarify our framework's scope and contributions:**
>
> **Important context**: Even in human cognitive science, there is currently no unified method to quantitatively measure cognitive load or working memory capacity that reliably predicts real-world task performance. Instead, cognitive load theory principles provide guidance on how to cope with these limitations. This is primarily due to complex interactions between internal mechanisms and various confounding factors. CLT itself is a qualitative framework, the components in CLT cannot be fully quantified in isolation.
>
>
> **Our work's contribution**: We focus on **establishing and providing initial validation** of a proof-of-concept framework showing that CLT principles can effectively guide LLM system design. We demonstrate validated proxies for design-time analysis and show how to leverage LLMs' emergent behaviors to operationalize cognitive principles. The complete "LLM Cognitive Science" research agenda—including runtime prediction and adaptive systems—represents important future directions that our foundational work enables.
>
> We next clarify: (1) fundamental issues preventing direct quantification in LLMs, (2) our solution approach, and (3) how our framework envisions future directions.

---

> ### Author Response · Authors · 2025-11-17
>
> ### **Three Fundamental Issues Preventing Direct Quantification:**
>
> #### **Issue 1: Closed-Source Models - We Utilize Emergent Behavior**
>
> For closed-source models, internal parameters (attention weights, hidden states) are inaccessible. Therefore, to make our framework more general, we **utilize LLMs' emergent social behaviors**—their ability to form collective minds to distribute the cognitive load—to apply CLT principles through a model-agnostic approach.
>
>
>
> #### **Issue 2: Working Memory - Capacity Alone Doesn't Predict Ability**
>
> Several recent papers have measured LLMs' working memory by mirroring classic human experiments (e.g. N-back tasks)[1]. However, these measurements show limited practical value for predicting real task performance.
>
> **The core problem**: Working memory capacity refers to how many **elements** one can simultaneously hold and manipulate. However, what constitutes an "element" depends on one's **chunking/grouping ability**—how well its attention mechanism can encode in-context information into meaningful units. In cognitive science, this is known as **information chunking** [2,3]. Consider a human example: when remembering a phone number "415-555-0123", humans naturally chunk it as three groups (415, 555, 0123) rather than ten individual digits, occupying only 3 working memory slots instead of 10. Current research measures WM capacity in isolation, but practical task performance requires understanding **both** capacity **and** how LLMs organize information into representational units.
>
>
>
> #### **Issue 2a: Cognitive Load is Dynamic - Not a Task's Inherent Property**
>
>
> Cognitive load is **not** a fixed property of a task—it **dynamically changes** during reasoning, depending on the model and intermediate steps. Recent theoretical work converges on a key finding: **reasoning strategies transform attention patterns during inference**, creating more sparse attention pattern that affect processing difficulty [4,5,6]. Specifically, CoT introduces sparse sequential dependencies that restructure information flow [5], breaks compositional tasks into distinct processing phases [4], and creates coherent reasoning chains that integrate earlier steps [6]. This transformation means cognitive load varies with different models and reasoning strategies—making it impossible to predict CL before inference begins, as it emerges through the reasoning process itself.
>
>
> #### **Issue 2b: Metacognitive Sensitivity - The Confounding Factor**
>
>
> Even if we could measure CL and WM accurately, cognitive overload alone cannot predict performance outcomes without considering **metacognitive sensitivity**—the ability to accurately assess one's own cognitive state. When facing high cognitive load, models exhibit two divergent behaviors: (1) appropriately acknowledging its uncertainty to its response, or (2) confidently employing hallucination without recognizing their reasoning limitations. This mirrors findings from human cognition where metacognitive accuracy varies independently of actual ability [9], demonstratiing that both skilled and unskilled individuals show poor calibration between confidence and performance when cognitive overload. On LLM, recent approaches address this through attention-based uncertainty quantification [7] and RL-based uncertainty training [8]. Once metacognitive sensitivity is properly calibrated, our validated proxies can enable the runtime prediction systems. With properly calibrated metacognitive signals distinguishing true uncertainty from processing CL, we could determine whether an LLM is experiencing cognitive overload and thus **adaptively decide when to invoke multi-agent collaboration**—directly enabling the adaptive framework the reviewer describes (**addressing Q2**).
>
> ---

---

> ### Author Response · Authors · 2025-11-17
>
> ### **Our Solution: Validated Proxies + Emergent Behavior Application**
>
> We choose to use proxies to justify the human/LLM anology of working memory mechnism. And with these validted proxy, we lay the foundation for system design. This fully justify why we can further apply CLT principle in desinging better multi-agent framework.
>
> #### **Step 1: Establish Test-Time Proxies**
>
> We established two empirically validated proxies grounded in existing literature. These proxies require ground-truth answers, which is why they serve as **design-time validation tools** rather than runtime predictors. Attention Entropy is a proxy derived from Cognitive Load *definition* and Perplexity is a proxy derived from Cognitive Load *mechnism*.
>
> **Attention Entropy as a proxy for Intrinsic Cognitive Load**: To understand what attention entropy measures, we must first clarify the concept of working memory. In cognitive science, working memory refers to the capacity to simultaneously hold and manipulate a limited number of **elements** [15,16]. A high cognitive load task requires using more element simultaneously to solve. Analogous to how human attention, in LLMs, attention entropy directly measures how much "element" is used simultaneously for prediction. When attention is scattered uniformly across many tokens(high entropy), it suggests the model have to use all in context information simultaneously to predict the next token, corresponding to high cognitive load. Additionally, this interpretation is supported by recent theoretical work showing that  Chain-of-Thought create sparse attention patterns by introducing sequential dependencies that naturally chunk information and thus enhancing performance [4,5,6]
>
>
> **Perplexity as a proxy for Processing Difficulty**: In human cognition, metacognitive judgments serve as cues for processing difficulty assessment which faithfully reflects cognitive load [17] . Similarly, perplexity measures LLMs' comprehension difficulty by quantifying prediction uncertainty. Metacognitive theory establishes two distinct scenarios [10,11]: when not under **cognitive overload**, confidence signals accurately reflect task difficulty; under **cognitive overload**, retrieval quality degrades and uncertainty signals become saturated. This maps directly to LLMs—our framework validates perplexity responds differentially in our test-time experiemnt; And our framework also explain why perplexity fails to predict quality during inference [12].
>
> These proxies require ground-truth answers for validation, making them **design-time tools** rather than runtime predictors. They help us understand and validate CLT mechanisms in controlled settings, providing the foundation for system design.
>
>
>
> #### **Step 2: Apply CLT Principles via Emergent Behavior**
>
> Having established the CL/WM analogy in LLMs, we can now apply CLT principles to guide system design. We leverage LLMs' **emergent social behaviors** (thinking-style, collected minds) to operationalize CLT principles. **Transactive Memory System (TMS)** and **Communication Moderator (CM)** are established constructs from human team research [13,14]. Our framework translates these into computational modules: TMS for cognitive offloading via expertise tracking, CM for bounded interaction , and Thinking Style Orchestrator for dynamic role adaptation. These CLT-derived principles guide multi-agent collaboration to manage cognitive limitations. See Section 3 for architecture and Section 5 for empirical validation.
>
>
>
> ---

---

> ### Author Response · Authors · 2025-11-17
>
> ### **How Our Framework Envisions Future Directions**
>
>
> Our work lays the **initial foundation** for a comprehensive "LLM Cognitive Science" research agenda. To enable complete runtime adaptation, three key quantities require measurement: **(1) Working Memory with Chunking** - understanding how LLMs organize and chunk information into representational units, determining what constitutes an "element" in their working memory; **(2) Dynamic Cognitive Load During Inference** - tracking how CL transforms through reasoning chains, where attention-based runtime monitoring offers potential solutions; **(3) Metacognitive Sensitivity** - calibrating uncertainty signals to distinguish processing difficulty from knowledge gaps through RL-based training or attention-based quantification. Once these components are quantified, our validated proxy framework provides the foundation for runtime prediction systems that can adaptively manage cognitive load—directly addressing the reviewer's vision in Q1 and Q2. Moreover, this bidirectional potential extends to cognitive science itself—studying LLMs offers more controllable experimental paradigms than human studies, potentially revealing new insights about cognitive mechanisms.
>
> We are now adding the above discussion to Future Work section and also in the main paper to strengthen our frameworks' vision.

---

> ### Author Response · Authors · 2025-11-17
>
> ## **Clarification to W1: "Cognitive science is merely rhetorical framing...lacks true innovation"**
>
> We clarify that our cognitive science framework is not merely rhetorical framing but grounded in theoretical and empirical evidence, following the same scientific methodology by which human Cognitive Load Theory was originally established.
>
> **Our approach mirrors how CLT was discovered in human cognition**: Human cognitive scientists first observed that the brain can only attend to a limited amount of information at once—working memory [15,16]. They then developed Cognitive Load Theory to explain how managing this WM limitation enables effective learning and problem-solving, particularly in collaborative settings.
>
> **Establishing the same foundation in LLMs**: Our analogy begins with a foundational correspondence grounded in architectural constraints. Both human neural processing and LLMs' attention mechanism (softmax over all tokens, bounded by model dimensions) face bounded capacity for simultaneous information processing. We refer to this as "LLM's working memory" not as mere metaphor, but as a mechanistic parallel (Section 3.1). Specifically, our experiments demonstrate several reliable proxy for this working memory constraint.
>
> This validated framework provides explanatory power for numerous phenomena observed in recent literature. These connections demonstrate that our working memory perspective is not merely descriptive but offers a unifying theoretical lens for understanding diverse empirical findings in LLM behavior.
>
> **From validated principles to system design**: Just as CLT guides human collaboration, we demonstrate these principles can guide LLM multi-agent systems. **TMS** implements cognitive offloading via expertise tracking, **CM** manages group cognitive load through bounded communication (N=2-3, β=0.3), and **dynamic thinking styles** reduce extraneous load from rigid role adherence. These aren't arbitrary choices but principled applications of validated cognitive theory (Section 3.2).
>
> **Empirical validation confirms predictions**: Our framework makes testable predictions that **distinguish it from heuristic MAS**: certain tasks won't benefit from collaboration (coordination cost exceeds benefit), confirmed in Table 2; exceeding WM limits degrades performance, confirmed in Figure 4; cognitive offloading improves efficiency, validated through proxy measurements (Tables 5-6).
>
>
> ---
>
> ## **Response to W3: "Pilot study merely restates obvious facts"**
>
>  **We clarify that the pilot study reveals non-trivial findings:**
>
> **Experiment 2 (Perplexity):**
>
> Our finding reveals a **differentiated effect**: instructions help hard tasks but hurt easy tasks when prefilled with question and answer. This pattern demonstrates that additional information creates extraneous cognitive load for easy tasks where processing capacity is already sufficient, directly validating CLT's prediction about unnecessary information impairing performance.
>
> Why this isn't trivial: While "harder task -> higher perplexity" might seem obvious, this relationship **only holds in our controlled prefill setting**. Recent work [12] shows that during actual inference, perplexity has **almost no correlation** with task performance. This is because cognitive overload causes perplexity to saturate and lose its discriminative power. Thus, our experiment captures the CLT mechanism explaining both why our proxy validates CLT principles in controlled settings and why perplexity fails as a quality predictor during inference.
>
>
> ---
>
> **Experiment 1 (Attention Entropy) - Mathematical and Empirical Non-triviality**:
>
> Standard mathematical intuition suggests that as sequence length increases, attention entropy should increase. For a probability distribution over $N$ tokens with noramlized attention weights $a_i$ (i.i.d), attention entropy is $H = -\sum_{i=1}^{N} a_i \log a_i$. When N is large, $H\sim \log N$. Thus, longer sequences have higher attention entropy naturally. However, our empirical results show that adding reasoning steps -- attention entropy **decreases** (Table 6). This aligns with [5]'s validation that CoT produces sparse attention patterns—reasoning introduces sequential dependencies that structure information flow, creating focused attention despite longer sequences where sparsity dominates length effects.
>
> ---

---

> ### Author Response · Authors · 2025-11-17
>
> ## **Response to Q3: "How sensitive is the performance of CoThinker to the chosen communication topology?"**
>
> CoThinker's performance is highly sensitive to communication topology for **high cognitive load tasks** (math, data analysis), but robust for **low-load tasks** (instruction).
>
> **Key evidence:**
> - Math/logic Reasoning: Performance peaks at N=2-3, $\beta$=0.3-0.6, with ~20% swings outside optimal ranges
> - Instruction: <5% variation across all N and β values
>
> The small-world topology provides the right balance - sensitive enough to need tuning for hard tasks, but predictable in its patterns.
>
> ---
>
> **References:**
>
>
> [1] Zhang, Chunhui, et al. "Working memory identifies reasoning limits in language models." Proceedings of the 2024 Conference on Empirical Methods in Natural Language Processing. 2024.
>
> [2] Miller, George A. "The magical number seven, plus or minus two: Some limits on our capacity for processing information." Psychological review 63.2 (1956): 81.
>
> [3] Cowan, Nelson. "The magical number 4 in short-term memory: A reconsideration of mental storage capacity." Behavioral and brain sciences 24.1 (2001): 87-114.
>
> [4] Li, Yingcong, et al. "Dissecting chain-of-thought: Compositionality through in-context filtering and learning." Advances in Neural Information Processing Systems 36 (2023): 22021-22046.
>
> [5] Wen, Kaiyue, et al. "From Sparse Dependence to Sparse Attention: Unveiling How Chain-of-Thought Enhances Transformer Sample Efficiency." The Thirteenth International Conference on Learning Representations.
>
> [6] Yingqian Cui, Pengfei He, Xianfeng Tang, Qi He, Chen Luo, Jiliang Tang, Yue Xing, A Theoretical Understanding of Chain-of-Thought: Coherent Reasoning and Error-Aware Demonstration. Proceedings of The 28th International Conference on Artificial Intelligence and Statistics, PMLR 258:3475-3483, 2025.
>
>
> [7] Li, Y., et al. (2025). Language Model Uncertainty Quantification with Attention Chain. arXiv:2503.19168v2
>
> [8] Damani, Mehul, et al. "Beyond binary rewards: Training lms to reason about their uncertainty." arXiv preprint arXiv:2507.16806 (2025).
>
> [9] Burson, K. A., Larrick, R. P., & Klayman, J. (2006). Skilled or unskilled, but still unaware of it: how perceptions of difficulty drive miscalibration in relative comparisons. Journal of Personality and Social Psychology, 90(1), 60-77.
>
> [10] Koriat, A. (2007). Metacognition and consciousness. In P. D. Zelazo, M. Moscovitch, & E. Thompson (Eds.), The Cambridge handbook of consciousness (pp. 289-325). Cambridge University Press.
>
> [11] Koriat, A. (2012). The feeling of knowing: Some sources of feelings of knowing and not knowing. In J. Dunlosky & S. K. Tauber (Eds.), The Oxford handbook of metamemory (pp. 213-287). Oxford University Press.
>
> [12] An, C., et al. (2024). What is Wrong with Perplexity for Long-context Language Modeling?
>
> [13] Zhang et al. "Exploring Collaboration Mechanisms for LLM Agents: A Social Psychology View." ACL, 2024.
>
> [14] Kirschner et al. "A cognitive load approach to collaborative learning." Educational Psychology Review, 2009.
>
> [15] Baddeley, A. (1992). Working memory. Science, 255(5044), 556-559.
>
> [16] Cowan, N. (2001). The magical number 4 in short-term memory: A reconsideration of mental storage capacity. Behavioral and Brain Sciences, 24(1), 87-114.
>
> [17] Alter, Adam L., and Daniel M. Oppenheimer. "Uniting the tribes of fluency to form a metacognitive nation." Personality and social psychology review 13.3 (2009): 219-235.

---

> > ### Author Response · Authors · 2025-11-25
> >
> > Dear Reviewer XuKf, thank you for your feedback. We have provided a detailed rebuttal addressing your points on the challenges of direct WM quantification, the validity of our proxies as design-time tools, and the non-trivial findings of our pilot study. Could you please confirm if our explanations regarding the scope of our contribution and the validity of our measurement approach have clarified your concerns?

---

> > > ### Author Response · Authors · 2025-11-26
> > >
> > > Dear Reviewer XuKf, We have updated the PDF to include our rebuttal discussions on WM/CL quantification challenges (W2), proxy validation (W3), and future directions. Could you please confirm if our reply addresses your concerns? Thank you for your feedback.

---

> > > > ### Comment · Reviewer_XuKf · 2025-11-26
> > > >
> > > > Thank you for the response, which resolves my concerns about the validation experiments/perplexity and attention, and partially addresses my concern about quantifying cognitive load.
> > > >
> > > > Based on the response, I believe we have reached a consensus that, because it is not quantifiable, currently cognitive load theory can only offer some high-level guidance for the design of LLM agents, while the design of the CoThinker system, such as role assignment, the communication bus and the small world topology, does not go beyond the current mainstream design paradigms of LLM multi-agent systems. Therefore, I see this paper more as using concepts from cognitive science to explain existing mainstream LLM multi-agent designs. I also find the discussion of future directions in the authors’ response relatively common, rather than being uniquely inspired by cognitive science.

---

> > > > > ### Author Response · Authors · 2025-11-27
> > > > >
> > > > > Thank you for your engagement in this discussion, which has helped us to clarify the unique contribution of our proposed research roadmap.
> > > > >
> > > > >
> > > > > Our future directions are not generic engineering but follow the specific research trajectory of modern Cognitive Science, moving from behavioral observation to internal mechanisms. **Our previously outlined directions, such as dynamic CL measurement and chunking quantification, are fundamentally rooted in specific cognitive constructs  and are distinct from generic MAS optimization.** Our paper establishes the critical **behavioral and functional foundations** (validating attention entropy as a proxy), which is the necessary prerequisite for this deeper analysis. With this bridge established, we will be able to now operationalize recent cognitive findings on dynamic resource allocation, investigating how attention heads functionally mirror these biological strategies to manage load. This creates a unique **co-moving** frontier where LLM interpretability is directly informed by modern neuroscience, using our established proxies to test how computational attention mechanisms mathematically mirror biological strategies.

---

### Official Review · Reviewer_j9Gu · 2025-10-26

**Soundness:** 2
**Presentation:** 3
**Contribution:** 2
**Rating:** 6
**Confidence:** 3

**Summary:**

This paper investigates the performance limitations of Large Language Models (LLMs) on complex, multi-faceted tasks through the lens of Cognitive Load Theory (CLT) from cognitive science. The proposed multi-agent framework CoThinker operationalizes CLT principles by distributing intrinsic cognitive load through agent specialization and managing transactional load via structured communication and a collective working memory. Experiments on LiveBench and CommonGen-Hard demonstrate improved performance over the baselines, especially on high cognitive load tasks.

**Strengths:**

1. The application of Cognitive Load Theory to explain LLM limitations is novel and insightful, bridging human intelligence and machine intelligence.
2. Clear system design of the proposed method. Each component—agent specialization, the transactive memory system, and the communication moderator—directly maps to established principles for managing cognitive load in human collaborative systems.

**Weaknesses:**

1. Comparisons in experiments omit some strong structured reasoning and multi-agent baselines (e.g., Tree-of-Agents, Agents with a leader, etc.). Statistical significance and variance across seeds are not consistently reported.
2. Though each part of the system can be mapped to CLT, however, these three parts are typical settings for multi-agent systems. The inspiration from CLT to design the detail algorithms of each part is lacked.
3. There is scalability analysis in the manuscript, but the experiment sets are too few to completely show the correlations between the number of agents and the performance.

**Questions:**

1. Can you provide quantitative confirmation of small-world properties in the communication graph?
2. How sensitive are results to the choice of N and β beyond the reported ranges? Could you include task-wise adaptive selection strategies?
3. Can you provide more insights from CLT to explain the details of the system design?

---

> ### Author Response · Authors · 2025-11-21
>
> ## **Response to W1: Experimental Rigor and Baseline Selection**
>
> ### **Statistical Robustness**
>
> We use greedy decoding (temperature=0) for all refinement rounds. Initial generation uses temperature=0.25 to create diverse agent starting points, but subsequent refinement is deterministic. Multiple seeds would highly similar results under greedy decoding. Instead we choose to report the bootstrap variance representing instance wise variance under large samples.
>
> To demonstrate statistical power, we computed bootstrap variance for all methods on each test.
>
>
> **(1) Bootstrap standard errors**:
>
> We computed bootstrap standard errors for our main experiment, here we show partial results.
>
> | Model | Method | Math | Reasoning | Language | Instruction | Overall |
> |-|-|-|-|-|-|-|
> | Gemini-1.5-Flash-8B | IO | 1.70 | 2.92 | 2.26 | 1.61 | 1.13 |
> | | CoT | 1.77 | 2.89 | 2.15 | 1.65 | 1.14 |
> | | DMAD | 1.73 | 2.91 | 2.39 | 1.76 | 1.13 |
> | | CoThinker | 1.94 | 3.13 | 3.92 | 1.62 | 1.13 |
>
> We computed bootstrap standard errors across all method comparisons, showing all details for Table 8:
>
> | Model | Method | Overall (SE) | Math (SE) | Reasoning (SE) |
> |-|-|-|-|-|
> | Gemini-2.5-Flash | CoThinker | 76.3 (2.67) | 77.3 (2.57) | 75.6 (4.15) |
> | | DMAD | 65.9 (3.32) | 63.8 (4.48) | 67.4 (4.70) |
> | | IO | 51.9 (3.40) | 66.4 (3.78) | 41.5 (4.86) |
> | GPT-4.1-Mini | CoThinker | 66.0 (2.92) | 51.6 (3.72) | 76.4 (3.90) |
> | | DMAD | 49.3 (3.21) | 46.8 (3.84) | 51.0 (4.82) |
> | | IO | 47.6 (3.27) | 47.9 (3.91) | 47.4 (4.82) |
> | Qwen3-32B | CoThinker | 27.6 (2.89) | 23.3 (3.52) | 30.6 (4.22) |
> | | DMAD | 14.8 (2.38) | 17.1 (3.53) | 13.1 (3.25) |
> | | IO | 11.7 (2.33) | 12.8 (3.53) | 11.0 (3.14) |
> | DeepSeek-R1-8B | CoThinker | 13.6 (2.28) | 4.2 (1.42) | 20.4 (3.58) |
> | | DMAD | 8.3 (1.71) | 5.7 (2.00) | 10.2 (2.58) |
> | | IO | 4.8 (1.46) | 3.8 (1.94) | 5.4 (2.00) |
>
> The small SE values (1.4-4.9% across tasks) indicate tight confidence intervals. Statistical significance analysis: We computed 95% confidence intervals for several key comparisons:
>
> - Gemini-2.5-Flash on Reasoning (high-CL):  Non-overlapping CIs confirm CoThinker significantly outperforms IO (p<0.01).
> - GPT-4.1-Mini on Reasoning (high-CL): CoThinker significantly outperforms both (p<0.01).
> - DeepSeek-R1-8B on Reasoning (high-CL):  Non-overlapping CIs confirm significant gains (p<0.01).
>
>
> ### **Baseline Selection: Complementary Not Competing**
>
> CoThinker is not a competing methods to existing frameworks. It provides plug-and-play CLT principles that can enhance other architectures.
>
> Our baselines (MAD, DMAD) validate CLT-guided design in distributed peer-to-peer settings. Hierarchical frameworks like MetaGPT [2] use different coordination paradigms. These are complementary. Our validated CLT principle can also enhance hierarchical systems:
>
> (1) Our proposed CoThinker can directly be considered as a node in those hierarchical systems as a functional node [3].
>
> (2) Using our validated CLT principle we can restrict the in-degree of each tree node or use more capable agent at root node to coordinate cognitive load for hierarchical systems.
>
> (3) Our small-world implementation can also be considered in creating cross branch communication in a hierarchical tree structure to mange CL in this heterogeneous system.
>
> We focus on homogeneous distributed settings to isolate and validate CLT principles. Extension to heterogeneous hierarchical systems is future work our foundations enable. Our baselines validate our core claim: CLT provides principled guidance that outperforms heuristic designs.
>
> ---
>
> ## **Response to W2: How CLT Generates Principled Design**
>
> CLT's contribution is not inventing modules but providing CLT principled guidance for better design with theoretical justification.
>
> Multi-agent systems face questions where CLT provides theoretical grounding from cognitive science research on managing working memory in collaboration [3,4].
>
> ### **Key Design Dimensions**
>
> **(1) Communication architecture**: Traditional systems use all-to-all communication (high load) or fixed sparse connections. Our Communication Moderator implements small-world topology based on these principle (a) Working memory research showing a limited number of coordination sources prevent cognitive overloading [3,4]; (b) Diversity-bandwidth trade-offs [5]—small-world structure achieves local clustering (reducing integration cost -- consensus) and short paths (enabling information flow) [6].
>
> **(2) Dynamic roles**: Traditional systems use fixed roles requiring persona consistency[7]. Our Thinking Style Orchestrator generates task-adaptive styles, pushing agent towards diverse mental sets.
>
> **(3) Structured memory**: Flat history mixed up information. Our Transactive Memory System tracks "who knows what," as structured expertise tracking reduces redundant processing and enables cognitive offloading [8].

---

> ### Author Response · Authors · 2025-11-21
>
> ---
>
> ### **Quantitative Validation of Small-World Properties (Q1)**
>
> We computed small-world coefficients $\sigma$ for our networks, we use weighted graph with $1/w$ as distance metric. Results on math tasks (Gemini-1.5-Flash-8B):
> - M=6 N=3 beta=0.3: median $\sigma$=2.75
> - M=12 N=3 beta=0.3: median $\sigma$=2.87
> - M=18 N=3 beta=0.3: median $\sigma$=3.12
>
> Values $\sigma>1$ indicate small-world structure [1]. Our results show robust small-world properties across configurations, also showing more models with slightly higher small world property.
>
> | Model | Task | Median| $\sigma$ |
> |-|-|-|-------|
> | Gemini-1.5-Flash-8B | math | 2.670 |
> | DeepSeek-R1-8B | math | 3.285 |
> | Qwen3-32B | math |  3.041 |
> | Gemini-2.5-Flash | math |2.865 |
> | GPT-4.1-Mini | math |  2.725 |
>
>
> All model shows the small-world property under our methods.
>
> ## **Response to Q2**
>
> CoThinker's performance is highly sensitive to communication topology for high cognitive load tasks (math, data analysis), and less for low-load tasks (instruction).
> Key evidence:
> - Math/logic Reasoning: Performance peaks at N=2-3, beta=0.3-0.6.
> - Instruction: small variation across all N and β values
>
> For task-wise adaptive selection strategy, larger ranging probing is beneficial for more capable agent on high cognitive load tasks, particularly the reference number. For low cognitive load tasks, no need for a comprehensive probing as parameter are less sensitive and no much gains.

---

> > ### Author Response · Authors · 2025-11-21
> >
> > ## **Response to Q3: More Insights from CLT to System Design Details**
> >
> > We provide deeper connections between specific CLT principles, their manifestation in human cognition, and their operationalization in CoThinker.
> >
> > ### **From Working Memory Limits to Communication Constraints**
> >
> > Cognitive science has established that working memory can actively process only several information chunks simultaneously [3,4]. When processing demands exceed this capacity, performance degrades not from the task's inherent difficulty but from **extraneous cognitive load**—mental effort wasted on managing overwhelming information rather than solving the problem [7]. In collaborative settings, this manifests as information overload when team members must integrate too many simultaneous communication channels. In human, 2-3 active information sources per person often show better result [9].
> >
> > Our Communication Moderator directly operationalizes this principle through fixed in-degree, maintaining total information load. This design choice reflects a fundamental constraint rather than an engineering preference.
> >
> > ### **From Extraneous-Intrinsic Load Trade-offs to Small-World Topology**
> >
> > Cognitive Load Theory distinguishes between intrinsic load (inherent task complexity) and extraneous load (unnecessary mental effort from information presentation or coordination) [7]. Distributing intrinsic load requires diverse perspectives, but processing diverse information creates extraneous integration costs. Research on organizational networks reveals that small-world structures naturally resolve this tension [5,6]. High clustering allows similar experts to form cohesive groups where shared mental models reduce explanation overhead. Simultaneously, sparse long-range connections ensure diverse insights propagate quickly without overwhelming individual members with unfamiliar perspectives.
> >
> > Our implementation uses Watts-Strogatz (small world implementation) rewiring to create this balance. Starting from a initial network maximizing local clustering, we randomly rewire some edges. This produces local clusters where agents refine similar ideas with low integration cost, while sparse long-range connections distribute intrinsic load by introducing diverse viewpoints.
> >
> > ### **From Role Rigidity to Adaptive Cognitive Styles**
> >
> > Fixed role assignments in human teams—such as designating one person as "the critic" or "the implementer"—can become sources of unnecessary cognitive burden when roles lose relevance to current task demands [7]. Maintaining persona consistency across varied problem contexts consumes mental resources that could otherwise be devoted to actual problem-solving. This represents a form of extraneous cognitive load: effort spent on task-irrelevant activities (sticking to the role). Studies of effective human teams show that complex problems require adaptive specialization where members fluidly shift cognitive focus as task demands evolve [10], contrasting with rigid roles that work only for routine, predictable tasks.
> >
> > Our Thinking Style Orchestrator addresses this by generating cognitive approaches tailored to each specific task. Based on established thinking style frameworks [11]. This approach eliminates the mismatch problem of predefined roles and also reduce the cognitive load imposed by sticking to the role.
> >
> > ### **From Implicit Team Knowledge to Explicit Transactive Memory**
> >
> > Research on human teams discovered the Transactive Memory System (TMS)—shared awareness of "who knows what" that enables cognitive offloading [8]. Rather than each team member remembering all information, they maintain meta-knowledge about expertise distribution.
> >
> > We explicitly reinforce this mechanism in LLMs. Our TMS maintains structured records of expertise distribution and collective consensus. The TMS structure includes global consensus summaries, expertise mappings, and divergent views, thus enabling effective cognitive offloading
> >
> >
> >
> >
> > ---
> >
> > ## **References**
> >
> > [1] Humphries & Gurney (2008). Network 'small-world-ness': A quantitative method.
> >
> > [2] Hong et al. (2023). MetaGPT: Meta Programming for Multi-Agent Collaborative Framework.
> >
> > [3] Cowan (2001). The magical number 4 in short-term memory.
> >
> > [4] Miller (1956). The magical number seven, plus or minus two.
> >
> > [5] Aral & Van Alstyne (2011). The diversity-bandwidth trade-off.
> >
> > [6] Watts & Strogatz (1998). Collective dynamics of 'small-world' networks.
> >
> > [7] Sweller (2011). Cognitive load theory.
> >
> > [8] Wegner (1987). Transactive memory: A contemporary analysis.
> >
> > [9] Almaatouq et al. (2020) "Adaptive social networks promote the wisdom of crowds" PNAS
> >
> > [10] Kirschner et al. (2018). Collaborative cognitive load theory: An integrative review.

---

> > > ### Author Response · Authors · 2025-11-25
> > >
> > > Dear Reviewer j9Gu, thank you for your thoughtful review. We have posted a comprehensive response covering your questions on statistical rigor, baseline selection, and design principles. We would appreciate it if you could let us know if these clarifications resolve your questions.

---

> > > > ### Author Response · Authors · 2025-11-26
> > > >
> > > > Dear Reviewer j9Gu, We have updated the PDF with bootstrap variance analysis (W1), expanded CLT design principles (W2), and small-world coefficient validation (Q1). Could you please confirm if our reply address your concerns? Thank you for your review.

---

### Official Review · Reviewer_esfh · 2025-11-01

**Soundness:** 3
**Presentation:** 3
**Contribution:** 2
**Rating:** 6
**Confidence:** 4

**Summary:**

- This work explains the performance ceiling of in-context learning by comparing LLM attention mechanism with human working memory, where LLM has the similar cognitive load theory as in cognitive science and can be measured by attention entropy and perplexity.

-  Based on the CLT of LLM and solutions to cognitive overload, the work introduce a multi-agent framework, CoThinker, consisting of agent parallel thinking, transacutive memory system and communication moderator. The experiments show the effectiveness of this framework in improving LLM's performance in complex tasks.

**Strengths:**

- Cognitive Load Theory provides an explanation for LLM performance limits and a clear design rationale for multi-agent collaboration.

- CoThinker takes cognitive science principles—like working memory, collective cognition, and small-world communication—and turns them into practical, easy-to-understand tools. This connects human cognitive theory with machine collaboration.

- The authors tested their theory using quantitative measures (entropy, perplexity) and multiple benchmarks across different model families, showing it’s robust and works broadly. Detailed ablation studies (on communication moderators, TMS, and thinking styles) help figure out which mechanisms do the most to cut down cognitive load.

**Weaknesses:**

- CoThinker underperforms on low-load tasks (e.g., instruction following) due to communication overhead—suggesting inefficiency in simple contexts.

- The chosen proxies for cognitive load (attention entropy, perplexity) are suggestive but indirect. Can you give more explanation on the relationship between attention entropy and cognitive load?

**Questions:**

Why were attention entropy and perplexity chosen as cognitive load proxies, and how might alternative measures, e.g., gradient variance?

---

> ### Author Response · Authors · 2025-11-20
>
> Thank you for your thoughtful and constructive review.
>
> ### **1. Clarification to W1: Underperformance on Low-Load Tasks**
>
> We clarify that the underperformance on low-load tasks (e.g., Instruction Following) is **not a system failure, but a confirmation of the "Redundancy Effect" in CLT**.
>
> Cognitive Load Theory posits that collaboration imposes **transactional costs** (extraneous load) due to the need for communication and coordination [1].
> *   **For high-load tasks** (e.g., Math, Reasoning), the benefit of distributing the *intrinsic load* across agents outweighs this transactional cost, resulting in net performance gains.
> *   **For low-load tasks**, a single agent's working memory is sufficient. Introducing collaboration adds transactional costs without alleviating any bottleneck. Therefore, CLT predicts the performance with unnecessary coordination overhead.

---

> > ### Author Response · Authors · 2025-11-20
> >
> > ### **2. Response to W2 & Q1: Justification of Proxies and Alternatives**
> >
> > We clarify why **Attention Entropy** and **Perplexity** are the most appropriate proxies for measuring **Cognitive Load (CL)** with both theoretical and empirical justification, since they are derived directly from the definition and mechanism of CL (i.e. Attention Entropy proxy is derived from definition of CL, and Perplexity proxy is derived from mechanism of CL)
> >
> > **Why Attention Entropy? (Measuring "Information Chunks"):**
> >
> > To start, we first restate our analogy for Working Memory (WM) and Cognitive Load (CL). Our analogy starts with that human brain can only attend to a limited amount of information at once, a feature known as working memory (Sec. 3.1). More specific, in cognitive science, working memory capacity refers to how many **elements** one can simultaneously hold and manipulate; these elements are often referred to as "information chunks" [2, 3]. A high **cognitive load** task requires more working memory as it demands the simultaneous use of more elements to solve. Therefore, the CL of a task can be reflected by how many "information chunks" are needed to solve it.
> >
> > The architecture of LLM, due to its inherent attention mechanism, similarly restricts the amount of information it can focus on at once, where we refer this LLM feature as LLM's working memory. LLM predicts the next tokens through its attention mechanism, allowing the model to selectively focus on specific, relevant parts of the input context to generate a response.
> >
> > |Concept|Cognitive Science Concept|LLM Agent Mechanism|
> > |:-|:-|:-|
> > |**Storage**| Long-Term Memory (LTM)  | Parametric Knowledge (Weights/FFNs) |
> > |**Processing**|Working Memory (WM)| Attention Mechanism|
> > |**Load Metric**   | Element Interactivity   | Attention Entropy (Attention Sparsity)|
> >
> > By definition, we can measure how many "information chunks" are being active considered to complete a task. Thus, using Attention Entropy to measure the sparsity of information integration is a direct proxy:
> >
> > - Low Entropy = Sparse/Focused attention = The model only needs few "elements" to predict the answer (Low Load).
> > - High Entropy = Uniform attention = The model is attending to many distinct "elements" simultaneously to predict the answer (High Load).
> >
> > This analogy is justified through both our experiments and recent theoretical work. In our work, we find attention entropy correlates with task complexity (Table 1); also we find adding reasoning effectively reduce attention entropy even with longer context (Table 6). Our analogy also match recent work in theoretically explain why chain of thoughts work [4]; they also find COT is creating more sparse attention; in terms of our framework, it is that reasoning is creating cognitive offloading, allowing the model to process fewer chunks during solving the tasks.
> >
> > **Why Perplexity? (Measuring Processing Fluency):**
> >
> > We clarify that Perplexity serves as a proxy for **processing fluency**, representing the cognitive ease of processing information. Metacognitive research establishes that processing fluency (confidence) correlates with task difficulty and can reflect cognitive load faithfully when the subject is not under cognitive overload [5,6,7]. In our pilot study, we specifically measure perplexity on prefilled ground-truth answers and validate it. In contrast, during open generation, cognitive overload causes models to lose metacognitive calibration, often resulting in low-perplexity hallucinations where confidence fails to predict performance [8]. Both LLM phenomenon aligns with the cognitive science findings.
> >
> > **Alternatives: Gradient Variance**
> >
> > To better answer your question, we first clarify two concepts based on cognitive science. Human cognition relies on two distinct memory systems: **Long-Term Memory (LTM)** for storing vast knowledge, and **Working Memory (WM)** for actively processing limited current information [9]. Under our framework's analogy, an LLM's **Parametric Knowledge** (weights) corresponds to LTM, while the **Attention Mechanism** processing in-context information corresponds to WM.
> >
> > With this distinction in mind, we analyze Gradient Variance:
> >
> > -   **Gradient Variance over Parameters:** This measures the sensitivity of the model's weights (LTM) to data. It reflects how a model understand specific knowledge, but it does not capture the dynamic burden on the **contextual processing mechanism** (WM) during inference.
> > -   **Gradient Variance over Inputs:** We agree this is a promising alternative metric. It aligns with the concept of "Probing Behavioral Uncertainty" [10], where high sensitivity to input perturbations indicates that the model lacks a stable reasoning path. In this sense, it serves as a strong "white-box" alternative to **Perplexity** for measuring the model's **Cognitive Load**.

---

> > > ### Author Response · Authors · 2025-11-20
> > >
> > > **Summary of Metric Selection:**
> > > 1.  **To measure Element Interactivity by CL definition:** We currently find no better proxy than probing **Attention Patterns** (Entropy), as this directly quantifies how many "information chunks" are active in the WM "workspace." Future improvements might involve analyzing cross-layer attention causality to trace exactly how elements interact.
> > > 2.  **To measure through mechanism of cognitive load:** Apart from Perplexity, **Gradient Variance over Inputs** or **Direct Elicitation** (asking the model, potentially enabled by recent RL-based uncertainty training [11]) are indeed excellent alternatives probing out models perceived cognitive load.
> > >
> > > ---
> > >
> > > **References:**
> > >
> > > [1] Kirschner, F., et al. "A cognitive load approach to collaborative learning." Educational Psychology Review (2009).
> > >
> > > [2] Miller, George A. "The magical number seven, plus or minus two: Some limits on our capacity for processing information." Psychological Review (1956).
> > >
> > > [3] Cowan, Nelson. "The magical number 4 in short-term memory: A reconsideration of mental storage capacity." Behavioral and Brain Sciences (2001).
> > >
> > > [4] Wen, Kaiyue, et al. "From Sparse Dependence to Sparse Attention: Unveiling How Chain-of-Thought Enhances Transformer Sample Efficiency." ICLR (2025).
> > >
> > > [5] Koriat, A. "Metacognition and consciousness." The Cambridge handbook of consciousness (2007).
> > >
> > > [6] Koriat, A. "The feeling of knowing: Some sources of feelings of knowing and not knowing." The Oxford handbook of metamemory (2012).
> > >
> > > [7] Alter, Adam L., and Daniel M. Oppenheimer. "Uniting the tribes of fluency to form a metacognitive nation." Personality and social psychology review (2009).
> > >
> > > [8] An, C., et al. "What is Wrong with Perplexity for Long-context Language Modeling?" arXiv preprint (2024).
> > >
> > > [9] Baddeley, A. "Working memory." Science (1992).
> > >
> > > [10] Tanneru, Sree Harsha, et al. "Quantifying uncertainty in natural language explanations of large language models." arXiv preprint arXiv:2311.03533 (2023).
> > >
> > > [11] Damani, Mehul, et al. "Beyond binary rewards: Training lms to reason about their uncertainty." arXiv preprint arXiv:2507.16806 (2025).

---

> > > > ### Author Response · Authors · 2025-11-25
> > > >
> > > > Dear Reviewer esfh, thank you again for your constructive feedback. We have provided detailed responses addressing your main concerns, specifically explaining the underperformance on low-load tasks as the "Redundancy Effect" predicted by CLT, and clarifying the theoretical justification for using Attention Entropy and Perplexity as cognitive load proxies. Could you please confirm if these responses have sufficiently addressed your concerns?

---

> > > > > ### Author Response · Authors · 2025-11-26
> > > > >
> > > > > Dear Reviewer esfh, We have updated the PDF to incorporate our rebuttal discussions on the Redundancy Effect (W1) and proxy justification (W2, Q1). Could you please confirm if our reply resolves your concerns? Thank you for your feedback.

---

### Author Response · Authors · 2025-11-29
**Meta-Response**

We thank all reviewers for your constructive engagement. Our paper makes a two-fold contribution:

(1) establishing a validated LLM-Cognitive Load Theory (CLT) analogy through both theoretical grounding and empirical proxies (attention entropy and perplexity), and (2) demonstrating how Cognitive Load Theory (CLT) principles can guide multi-agent system design that outperforms heuristic baselines.

### ***Recognized Strengths***

Reviewers acknowledged our work's:
- **Novel theoretical bridge**: CLT provides "an explanation for LLM performance limits and a clear design rationale for multi-agent collaboration" *[esfh]*; the application is "novel and insightful, bridging human intelligence and machine intelligence" *[j9Gu]*; "interesting, and has the potential to open up new perspectives in understanding and improving large language models" *[Vnt8]*
- **Principled system design**: "Each component—agent specialization, the transactive memory system, and the communication moderator—directly maps to established principles for managing cognitive load" *[j9Gu]*; "connects human cognitive theory with machine collaboration" *[esfh]*
- **Comprehensive experiments**: "Robust and works broadly across different model families" *[esfh]*; "comprehensive, with solid ablation studies and well-organized analyses" *[XuKf]*; "generalizes well on different LLMs" *[Vnt8]*
- **Clear presentation**: "Clearly written and easy to follow" *[Vnt8]*; "good presentation" *[j9Gu,esfh]*



### ***Main Concerns and Resolutions***



All reviewers' primary concerns centered on **clarification of theoretical foundations and experimental details**, which we have comprehensively addressed:

1. **Theoretical foundation and proxy validity**: We clarified the distinction between cognitive load (internal processing state) and task difficulty (external property) through concrete examples *[Vnt8,XuKf]*. We justified our proxy choices by grounding them in established cognitive science literature on working memory and metacognition *[esfh,XuKf]*. We also demonstrated that our pilot study findings are non-trivial—e.g., reasoning decreases attention entropy despite longer context, contradicting intuitive expectation *[XuKf]*.

2. **CLT-derived design principles**: We explained how CLT generates specific, non-arbitrary design choices (e.g., bounded in-degree N=2-3, small-world topology) by mapping each component to established cognitive science constructs on collaborative cognition [j9Gu, XuKf]. We clarified that underperformance on low-load tasks confirms CLT's "Redundancy Effect" prediction rather than being a system failure *[esfh]*.

3. **Experimental rigor**: We provided bootstrap variance analysis, small-world coefficient validation, and parameter sensitivity analysis showing performance peaks for different tasks *[j9Gu]*. We showed substantial gains on newer models and clarified our baseline DMAD already incorporates principles from suggested related works *[Vnt8]* which we also rewrite our related work section.

These issues arose because the main text was constrained by page limits and could not fully elaborate these points; we have reorganized the manuscript and added the relevant clarifications in the revised version.

### ***Reviewer Confirmations***

- **Reviewer *XuKf*** (rating 4) confirmed our response "resolves my concerns about the validation experiments/perplexity and attention." Their remaining concern is a methodological perspective mindset difference—we follow the same procedure by which CLT was established in cognitive science (observation -> proxy validation -> principle application), which they view as post-hoc justification.

- **Reviewer *Vnt8*** (rating 2) confirmed they "will raise score to reflect" our clarification in our rebuttal. Their remaining concern about "overclaiming" is a writing issue we addressed in revision by scoping our claims more precisely.

- **Reviewers *esfh* and *j9Gu*** (both rating 6) received responses to all questions which we also updated the PDF incorporating rebuttal discussions.

Finally, all main shared concerns have been confirmed as resolved by the two replied reviewers.

---

### Note · Program_Chairs · 2026-01-17
**Submission Desk Rejected by Program Chairs**

The following references in this submission do not refer to real documents and/or have major errors in bibliographic information:

 Xuezhi Wen, Deepak Ganapathy, et al. Chain-of-thought reasoning without prompting. arXiv preprint, 2025a.